# ACTIVATION-DEACTIVATION: A GENERAL FRAMEWORK FOR ROBUST POST-HOC EXPLAINABLE AI

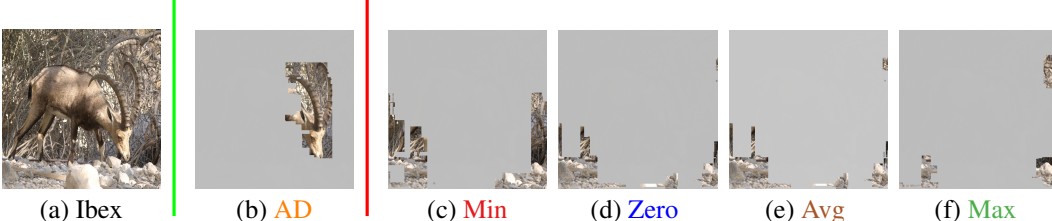

|        |        |         |          |         |         |
|--------|--------|---------|----------|---------|---------|
| (a) Ibex | (b) AD | (c) Min | (d) Zero | (e) Avg | (f) Max |

Figure 1: An image of an ibex (a) and its explanations using our method AD (b) compared to occlusion-based methods with different occlusion values (c)-(f) on the EfficientNet-v2 model.

## ABSTRACT

Black-box explainability methods are popular tools for explaining the decisions of image classifiers. A major drawback of these tools is their reliance on mutants obtained by occluding parts of the input, leading to out-of-distribution images. This raises doubts about the quality of the explanations. Moreover, choosing an appropriate occlusion value often requires domain knowledge. In this paper we introduce a novel forward-pass paradigm Activation-Deactivation (AD), which removes the effects of occluded input features from the model's decision-making by switching off the parts of the model that correspond to the occlusions. We introduce CONVAD, a drop-in mechanism that can be easily added to any trained Convolutional Neural Network (CNN), and which implements the AD paradigm. This leads to more robust explanations without any additional training or fine-tuning. We prove that CONVAD mechanism does not change the decision-making process of the network under regular inference. We provide experimental evaluation across several datasets and model architectures. We compare the quality of AD-explanations with explanations achieved using a set of masking values, using the proxies of robustness, size, and confidence drop-off. We observe a consistent improvement in robustness of AD explanations (up to $62.5\%$) compared to explanations obtained with occlusions, demonstrating that CONVAD extracts more robust explanations without the need for domain knowledge.

## 1 INTRODUCTION

Deep learning models are widely used in a variety of computer systems, including in mission-critical and safety-critical applications such as healthcare and autonomous driving. Due to the inherent opacity of these models' decision-making processes, there is an acute need for explanations of their decisions. Explanations are essential for tasks such as verification, planning, and medical diagnosis. A good explanation allows for increased confidence in the system, and for unearthing unexpected failure modes. A number of definitions of explanations have been introduced from various domains of computer science (Chajewska & Halpern, 1997; Gärdenfors, 1988; Pearl, 1988), philosophy (Hempel, 1965), and statistics (Salmon, 1989).

The existing explainability approaches for neural networks can be roughly divided into *white-box* and *black-box* techniques. White-box techniques are generally very efficient, however they require access to the internals of the model, which might not be possible and are typically architecture-dependent. Black-box techniques, in contrast, require only access to the model's inputs and outputs.

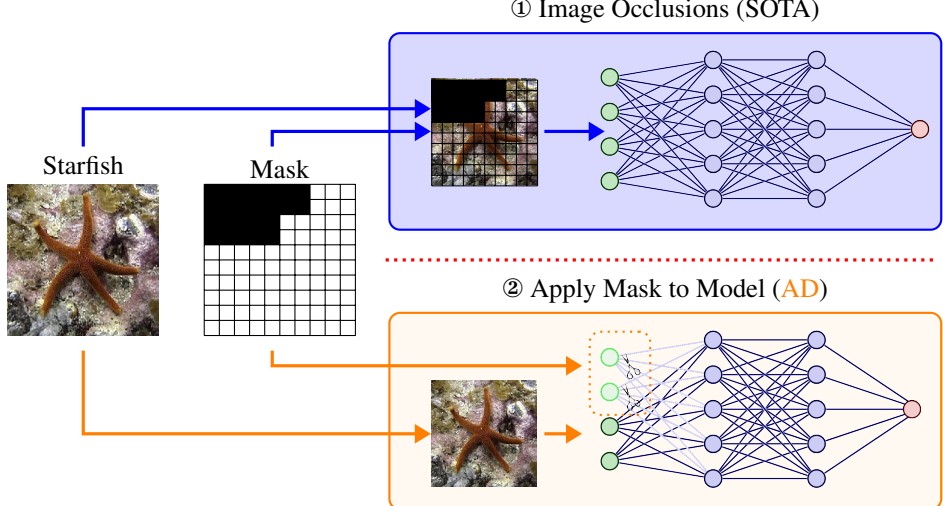

Figure 2: ① The state-of-the-art approach to generating perturbations for *post hoc* explainability: masking parts of the input image. ② Our approach Activation-Deactivation (AD): occluding parts of the *model*. AD preserves the spatial locality of the unmasked features while removing any consideration of the masked features, remaining *in distribution* for the input features to the model.

They typically make changes to the inputs in order to observe the effect on the output. However, recent work observes that mutations often result in out-of-distribution (*o.o.d.*) inputs, and the model's output on these does not accurately reflect the model's decision process (Hooker et al., 2019; Zheng et al., 2024). Moreover, choosing the right mutation (occlusion) values is key: if the occlusion value carries semantic meaning for the current model and current input dataset, the results might not reflect the model's reasoning on the original input. Therefore, in order to attain accurate explanations, domain-expertise and domain-specific adaptations are needed (Blake et al., 2025; Yang et al., 2026; Zhao et al., 2021; Botari et al., 2020; Meng et al., 2024; Sivill & Flach, 2022; Haunschmid et al., 2020; Mishra et al., 2017).

In this paper we introduce *Activation-Deactivation (*AD*)*, a novel paradigm that essentially forces the system to not take into account the occluded parts of the input, hence solving both the *o.o.d.* and the occlusion value problems. Intuitively, by only considering the unmasked parts of the model, we are utilizing a sub-network in the model. This sub-network exists within the original model and, hence, is in distribution over all decision paths through the model. The goal is to find a minimal sub-network that returns the original decision of the model on the original values of the input. Based on the definition of AD, we introduce a drop-in mechanism, CONVAD, that can be added to any CNN at test-time and which deactivates the set of activations that maps to the perturbation, effectively passing a cut-out of the unperturbed region to the model. To achieve this, we pass, as input to the model, the original example as well as the set of binary masks that is utilized by the XAI method to highlight which set of input features needs to perturbed. Figure 1 shows an example of an image labeled 'ibex' and an explanation obtained using AD and compares it with explanations obtained using state-of-the-art occlusion-based methods with different occlusion values. For this image of an ibex, the AD explanation clearly matches our intuition, as it consists of the ibex' head with its unique-looking horns. In contrast, explanations obtained using different common occlusion values show small areas of the background and contain almost no part of the ibex.

We implemented our framework and present experimental results on a number of CNN models and a number of datasets. Our results are over CNNs: it is straightforward, however, to extend AD to other architecture classes. Our empirical investigation evaluates CONVAD, along with other conventionally-used occlusion values, extracting explanations at different thresholds of confidence (with respect to the original confidence on the full input). We then test the robustness of each of the AD explanations by "planting" these explanations onto 100 different randomly selected colors and background images, calculating the percentage of the images that are correctly classified by the model. From each dataset, we use a sample of $N = 150$ and observe that CONVAD is consistently

among the best, and usually *the* best, of the considered methods. AD consistently outperforms the best masking values by 30-40% across all thresholds.

Due to the lack of space, all proofs, some background, and detailed experimental results are relegated to the appendix. The full set of results, the code, and the datasets are submitted as a part of the supplementary material.

## 2 BACKGROUND

We use the XAI tool ReX (Chockler et al., 2024) to generate all the explanations in this paper. ReX is a causal explainability tool which uses the definition of explanation provided for image classifiers by Chockler & Halpern (2024). Causal explanations have an advantage over other definitions of explanation in that they are formally defined, easily quantified and capable of capturing many aspects of an image (Kelly & Chockler, 2025). In Appendix B, we provide a primer on actual causality, providing the relevant definitions of *actual cause* and the *degree of responsibility* for *depth-2* causal models, representing black-box image classifiers (Chockler & Halpern, 2024). Roughly speaking, depth-2 causal models consist of an input layer and one internal node, computing an output of the classifier.

**Definition 1** (Causal explanation for depth-2 models). For a depth-2 causal model $M$, $\vec{X} = \vec{x}$ is an explanation of $O = o$ relative to a set of contexts $\mathcal{K}$, if the following conditions hold:

EXIC1.  $(M, \vec{u})[\vec{X} = \vec{x}] \models O = o$ for all $\vec{u} \in \mathcal{K}$.

EXIC2.  $\vec{X}$ is minimal; there is no strict subset $\vec{X}'$ of $\vec{X}$ such that $\vec{X}' = \vec{x}'$ satisfies EXIC1, where $\vec{x}'$ is the restriction of $\vec{x}$ to the variables in $\vec{X}'$.

EXIC3.  There exists a context $\vec{u}'' \in \mathcal{K}$ and a setting $\vec{x}''$ of $\vec{X}$, such that $(M, \vec{u}'') \models (\vec{X} = \vec{x}) \wedge (O = o)$ and $(M, \vec{u}'') \models [\vec{X} = \vec{x}''](O \neq o)$.

In the context of image classification, $\vec{X} = \vec{x}$ is a set of pixels, $\vec{u}$ is a setting of pixel values, and $O = o$ is the model's classification of the original image. $\mathcal{K}$ is a set of permutations of the original image. EXIC1 means that, whenever the pixels $\vec{X} = x$ are present in some setting $\vec{u}$, $O = o$ must hold (*i.e.* the classification must not change). EXIC2 is a minimality condition, stating that $\vec{X} = \vec{x}$ should not contain any excess variables. Without this clause, a causal explanation could trivially be the entire image. Finally, EXIC3 says there must be at least one setting where both $\vec{X} = \vec{x}$ and $O = o$ hold, in which changes to the values in $\vec{X}$ leads to $O \neq o$. Changing the values of the pixels in $\vec{X} = \vec{x}$ must be able to lead to a change in classification.

**Definition 2.** *(Robustness of an explanation)* $\vec{X} = \vec{x}$, denoted by $\rho$, is a fraction (or a percentage) of a given set of contexts $\mathcal{K}$ in which setting $\vec{X}$ to $\vec{x}$ results in $O = o$. In other words, an explanation $\vec{X} = \vec{x}$ for $O = o$ is $\rho$-robust if setting the subset of input features $\vec{X}$ to the values $\vec{x}$ leads to changing the classification to $o$ for $\rho$-fraction of the set of images.

An explanation with $\rho = 1$ means that it changes the output to $O = o$ in all the contexts in which it is placed. Likewise, an explanation with $\rho = 0$ is unable to change the classification of any of the contexts in which it is placed. It is still an explanation in its original context however.

## 3 THE ACTIVATION-DEACTIVATION (AD) FRAMEWORK

AD performs inference by forcibly *deactivating activations* at each layer of a model that pertain to occluded features. This halts the propagation of effects to the next layer of the model. In what follows, we assume the same model $\mathcal{N}$, which we omit from the notation. We introduce two new families of functions: position ($pos_i$) and position-attribution ($\Phi_i$), where $i$ is a layer of $\mathcal{N}$. For the ease of presentation, we describe the framework for two-dimensional inputs (like 2D images), but the exact same procedure works for any number of dimensions.

**Definition 3** (Position function ($pos_i$)). $pos_i(\mathbf{z}_{ab})$ is an inverse mapping from a position $(a, b)$ in the set of intermediate representations $\mathbf{z}$ to the set of previous layer features pertaining to $\mathbf{z}_{ab}$. It returns

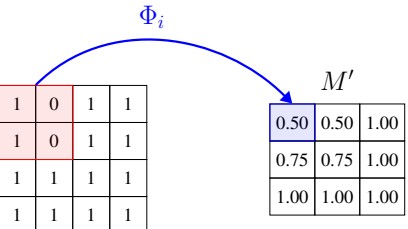

Figure 3: Visual demonstration of applying the position attribution function, in the highlighted section we have two unmasked values, leading to an attribution of 0.5.

the set of positions of elements from the previous layer in the set of all intermediate representations that map to $\mathbf{z}_{ab}$.

**Definition 4** (Position-Attribution function ($\Phi_i$)). The position attribution function $\Phi_i(\mathbf{z}_{ab}, M)$ for a given mask $M$ returns a ratio of masked to unmasked input features pertaining to $\mathbf{z}_{ab}$.

$$\Phi_i(\mathbf{z}_{ab}, M) = \frac{1}{|pos_i(\mathbf{z}_{ab})|} \sum_{pos_i(\mathbf{z}_{ab})} \mathbf{1}(M[k,l] > 0),$$

where $\mathbf{1}$ is the indicator function, and $M[k,l]$ is a cell in the mask $M$. 0 indicates that all the features in the current operation are masked, and 1 that they are all unmasked. In-between values represent varying ratios of masked and unmasked features. Again, similar definitions apply for $n$-dimensional inputs.

The hyperparameter $\tau$ is a threshold for $\Phi_i(\mathbf{z}_{ab}, M)$, below which sets of activations are discarded. Setting $\tau$ to 0 only deactivates fully masked regions; values above 0 deactivate some unmasked parts that border the masked regions. In Section 4, we discuss why $\tau > 0$ may be needed in certain cases to ameliorate the effect of leaks (see also Figure 4).

The output of $\Phi_i(\mathbf{z}_{ab}, M)$ is used to compute an updated mask $M'$ for the layer $i$ as follows:

$$M_i'[a, b] = \mathbf{1}(\Phi_i(\mathbf{z}_{ab}, M) > \tau).$$

Deactivation need not be performed at every step in the model. It is only performed prior to a layer resizing operation or an operation which averages or converts data including the masked values. The locations where we perform deactivation are called *checkpoints*.

We can now formally define Activation-Deactivation. Note that we use $\Phi_i(\cdot)$ to illustrate that position-attribution and thresholding have been performed over all $\mathbf{z}$.

**Definition 5** (Activation-Deactivation (AD)). Given a pair $(\mathbf{x}, M)$ of the input $\mathbf{x}$ and the accompanying binary mask $M \in \mathbb{R}^2$ and a Deep Neural Network (DNN) $\mathcal{N}$, the output, $O^{(i)}$, at a checkpoint after an intermediate layer $\mathcal{N}^{(i)}$ in a AD forward pass, is defined as:

$$O^{(i)} = \mathcal{N}_{(i)}(\mathbf{z}) \odot M_i'$$

Where $M_i'$ is the updated mask at the checkpoint $i$. In the first checkpoint we set $M_i' = M$. The AD output of a given layer is defined as the 2-tuple:

$$\text{AD}^{(i)} = (O^{(i)}, M_i')$$

**Putting the AD framework in the context of actual causality** The AD operation *removes* some of the inputs. Essentially, it constructs a smaller network, deactivating the neurons that map to parts of the input we wish to occlude.

Recall that we represent black-box DNNs as depth-2 causal models (Section 2). To capture the AD operation, we introduce *restricted* depth-2 causal models as follows.

**Definition 6** (Restricted depth-2 causal model). Given a depth-2 causal model $M_{\mathcal{N}}$ representing a black-box DNN $\mathcal{N}$ and a subset $\vec{V'} \subseteq \mathcal{V}$, a *restricted* causal model $M_{\mathcal{N}}|_{\vec{V'}}$ is derived from $M_{\mathcal{N}}$ by removing the input variables not in $\vec{V'}$ and restricting the function computed by the (single) internal node to the variables in $\vec{V'}$.

We note that in general, a restriction of a function to a subset of its parameters is not defined. However, as the model represents a DNN, we assume that it computes a *variadic* function, that is, a function that can accept a varying number of arguments. Hence restricting to a subset of the inputs results in a computable function.

Armed with the notion of restricted causal models, we define AD explanations as follows.

**Definition 7** (AD-explanation for depth-2 models). For a depth-2 causal model $M_\mathcal{N}$ computing a variadic function, $\vec{X} = \vec{x}$ is an AD-explanation of $O = o$ if it is a causal explanation of $O = o$ according to Definition 1 for all restrictions $M_\mathcal{N}|_{\vec{V'}}$ of $M_\mathcal{N}$ in which $O = o$ for the set $\mathcal{K}$ of all possible combinations of values of variables in $\mathcal{V}$ (which are all possible contexts for $M_\mathcal{N}$).

**Lemma 3.1.** AD-*explanations (Definition 7) are causal explanations (Definition 1) with respect to the set of all possible contexts.*

The proof follows from the observation that the set of all possible restrictions of the causal model includes the original model.

**Corollary 1.** *Causal explanations over the set of all possible contexts are prime-implicant explanations, we have that* AD-*explanations are also prime-implicant explanations (but the inverse does not necessarily hold).*

Following from Theorem 3.1, we know that AD-explanations are causal explanations. The proof that causal explanations are prime-implicant explanations but that the inverse is not necessarily true is provided in

# 4  THE CONVAD ALGORITHM

We implement the AD forward pass for a CNN in the algorithm CONVAD. CONVAD effectively converts a CNN to a variadic function, giving it the ability to handle variable sized inputs with the features to be considered being conveyed by the accompanying binary mask. This is achieved by systematically setting the values of intermediate representations pertaining to masked regions of the input to zero (deactivating) at strategic locations (CONVAD checkpoints) throughout the model. The checkpoints are locations where the current intermediate representations pertaining to masked regions will have downstream influence in the forthcoming set of intermediate representations unless their effects are removed in situ. CONVAD needs to accommodate two different types of checkpoints, namely: (1) $shape(\mathbf{z})$ is constant between $C^N$ and $C^{N+1}$, and (2) $shape(\mathbf{z})$ is altered between $C^N$ and $C^{N+1}$.

Case (1) is the scenario in which only shape-preserving operations have been performed since the last convolution operation. These are layers such as activation layers, dropout layers and normalization layers. In this case, the checkpoint is immediately after the last operation prior to the convolutional layer $C^{N+1}$. In case (2), *dimensionality-altering (DA)* operations have been performed prior to the convolutional layer $C^{N+1}$. This could be due to a parametric operation (i.e. interpolation) or an external additive or subtractive procedure (i.e. concatenation or deletion of representations). In this scenario, in addition to the checkpoint for case one, two additional checkpoints are added, one immediately prior to the DA operation and one immediately after. The checkpoint immediately prior ensures that we deactivate the effects of prior shape-preserving operations over the masked regions that may have occurred between $C^N$ and the DA operation.

The position-attribution function for the binary mask in case (1) is an identical convolution to $C^N$ with a mean-value kernel ($C_{ij} = \frac{1}{m \times n}$). This results in an identically-shaped representation as $\mathbf{z}$, where each entry is a scalar between 0 and 1, denoting the fraction of unmasked/masked features in the corresponding output of $C^N$. In case (2), the position-attribution function for a parametric dimension-altering operation is dependent on the operation being performed. In the case of external additive procedures, depending on whether the influence of the addition is to be considered, a set of 1s or 0s is added to the mask corresponding to the positions of the added representations. Similarly, for subtractive procedures, we simply remove the positions from the mask corresponding to the removed representations. Once the required representations and their masking values have been added/removed due to external procedures, we can simply convolve the updated mask with the mean-value kernel, similar to case (1), in order to get the position-attribution.

---

**Algorithm 1** CONVAD Forward Pass

---

**Input:** Input $x$, Binary Mask $M$, AD model $\mathcal{N}$
**Output:** Final layer output $\mathbf{z}$

1: $\mathbf{z} \leftarrow x$
2: **for** each layer $\mathcal{N}_{(i)} \in \mathcal{N}$ **do**
3:     $\mathbf{z} \leftarrow \mathcal{N}_{(i)}(\mathbf{z})$
4:     **if** $\mathcal{N}_{(i)}$ is conv **then**
5:         $\frac{J_{m \times n}}{m \times n} \leftarrow$ initialize kernel with $shape(\mathcal{N}_{(i)}^k)$ and all values set to $\frac{1}{m \times n}$
6:         $M \leftarrow \frac{J_{m \times n}}{m \times n} * M$
7:         $M \leftarrow M > \tau$
8:     **end if**
9:     **if** $\mathcal{N}_{(i)}$ is DA operation **then**
10:         $M \leftarrow \Phi_i(M)$
11:         $M \leftarrow M > \tau$
12:     **end if**
13:     **if** checkpoint **then**
14:         $\mathbf{z} \leftarrow \mathbf{z} \odot M$
15:     **end if**
16: **end for**
17: **return** $\mathbf{z}$

---

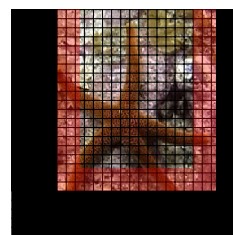 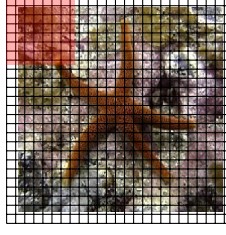 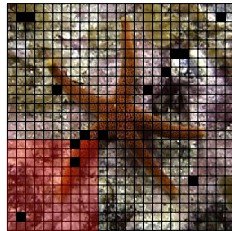

(a) Mixture of masked and unmasked regions

(b) Mixture of input features and padding

(c) Random perturbations

Figure 4: Scenarios where leaks can occur when applying CONVAD.

Hence, the CONVAD operation can be expressed as:

$$O^{(i)} = \begin{cases} \mathbf{z}_i \odot \mathbf{1}^{\oplus}(\sum(\frac{J_{m \times n}}{m \times n} * M_i) > \tau), & \text{if case 1} \\ \begin{cases} \mathbf{z}_i \odot \mathbf{1}^{\oplus}(\sum(\frac{J_{m \times n}}{m \times n} * M_i) > \tau) & \text{prior to DA} \\ \mathbf{z}_i \odot \mathbf{1}^{\oplus}(\Phi_i(M_i') > \tau)) & \text{post DA} \end{cases} & \text{if case 2} \end{cases} \quad (1)$$

Here, the $\oplus$ denotes that the updated mask is vectorized to $\mathbb{R}^{dim(\mathbf{z})}$ in order to apply the Hadamard product. $J_{m \times n}$ denotes a $m \times n$ matrix of ones. $M_i'$ is the updated mask after the checkpoint.

Given a CNN $\mathcal{N}$, we denote by $\mathcal{N}'$ the result of application of CONVAD on $\mathcal{N}$. The following theorem states that $\mathcal{N}'$ performs identically to $\mathcal{N}$ for images without occlusions, proving that CONVAD is a restriction of the original function and not a different function.

**Theorem 1.** *Given a model $\mathcal{N}$ and the result $\mathcal{N}'$ of applying CONVAD to $\mathcal{N}$, the output of $\mathcal{N}'$ is equal to the output of $\mathcal{N}$ on all inputs without occlusions (that is, without an accompanying binary mask).*

**Leakage Scenarios** As discussed above, CONVAD relies on a threshold value, $\tau$, that is used in order to determine if an activation is to be turned on or off. This is because it is impossible to separate the effects of masked and unmasked examples in certain edge cases and masking strategies. Figure 4 depicts three possible scenarios where the convolution operation straddles masked and unmasked regions of the input. In such scenarios, we are faced with two issues. **1)** for scenarios such as Figures 4a and 4b, the convolution window includes, in addition to an unmasked region, a portion

of either the masked region (4a) or the padding (4b). **2**) in Figure 4c, due to non-contiguous perturbation strategies, the convolution windows overlaps with each perturbations, making it impossible to determine which regions to deactivate.

# 5 EXPERIMENTAL RESULTS

In this section we outline our experimental setup and evaluation of AD-generated explanations compared to explanations obtained with a selection of masking values. All explanations are generated using the causal explainability tool REX (Chockler et al., 2024). REX is a black-box occlusion-based XAI tool computing an approximation of causal explanations (Definition 1). The exact computation of causal explanations is intractable even for depth-2 models (Chockler et al., 2024), thus for a given input $I$ REX computes approximate explanations over the set $\mathcal{K}$ of partial occlusions of $I$. Specifically, the explanations computed by REX satisfy EXIC1 and EXIC3 wrt this $\mathcal{K}$, but do not necessarily satisfy EXIC2, that is, they are approximately, but not guaranteed to be, minimal. Roughly speaking, REX first ranks features of the input according to their approximate degree of responsibility for the classification (see Appendix B) and then uses a greedy algorithm on the ranked list of features to extract approximate explanations. A key observation relevant for this paper is that REX uses partial occlusions of the input to construct $\mathcal{K}$, which it implements by replacing the occluded parts with occlusion values, and assumes that the occluded parts of the input do not affect the model's decisions. We repeat the robustness experiments using LIME with a smaller sample ($N = 50$) to replicate the results and demonstrate the efficacy of CONVAD using another black-box tool.

As proxy for the quality of explanations, we computed the following standard measures in all scenarios: **1**) robustness of explanation against solid-color backgrounds and independent and identically distributed (IID) examples; **2**) confidence of explanations; **3**) average size of explanations.

Our experiments were performed on 3 convolutional models: RegNetY-12GF (Radosavovic et al., 2020), ResNet-50 (He et al., 2016), and EfficientNet-V2 (Tan & Le, 2021), and on 4 standard datasets: ImageNet-1k (Deng et al., 2009), ImageNet-v2 (Recht et al., 2019), CalTech-256 (Griffin et al., 2022) and PASCAL-VOC (Everingham et al., 2010). The sets of models and datasets were selected in order to show the generality of the CONVAD mechanism and the quality of CONVAD explanations across a spectrum of architectures and inputs. The $\tau$ hyperparameter (Definition 4) is set to 0 for all experiments. For each image, we calculated a set of explanations by varying the *confidence threshold* $\gamma$ of the explanation as a multiplication factor of the model's confidence on the original example. We calculated explanations at $\gamma = 0, 0.1, 0.3, 0.5, 0.7$ and $0.9$ (that is, for $\gamma = 0.9$ we only consider explanations at confidence at least $0.9$ of the confidence of the model in the original classification).

## 5.1 ROBUSTNESS AGAINST SOLID-COLOR BACKGROUNDS

We measured $\rho$-robustness of explanations (Definition 2) as a proxy for their quality. We considered two sets of contexts. The set $\mathcal{K}_1$ consists of solid-colored images, hence $\rho$-robustness for $\mathcal{K}_1$ would indicate whether the model can identify the original class using the information available in the explanation only. The set $\mathcal{K}_2$ consists of images from a different class from the same dataset, highlighting the saliency of the features in the explanation with respect to the original input. In both classes we use 100 backgrounds against which to test each explanation.

Figure 5 demonstrates the results for AD, Min, Max, Avg and Zero for uninformative backgrounds (solid color). The results are reported for all thresholds $\gamma$. It is easy to see that in most cases the AD explanations are significantly more robust than all other masking strategies, and this is especially prominent for RegNetY-12GF and EfficientNet-V2 models. The results for the CalTech-256 dataset are more mixed at the lower confidence threshold. This can be attributed to the increased amount of noise in explanations at lower confidence levels. Figure 6 replicates the results using LIME, comparing Masking and AD, and we observe similar results. The reduction in robustness can be attributed to the crudeness of the saliency of LIME compared to REX.

An important observation is that there is *no consistent best masking value* for a given dataset or model. In RegNetY, we observe similar results for Min and Max for ImageNet-1k and ImageNet-1k_v2, however, in CalTech-256, using the Avg value seems to be the most appropriate strategy. In

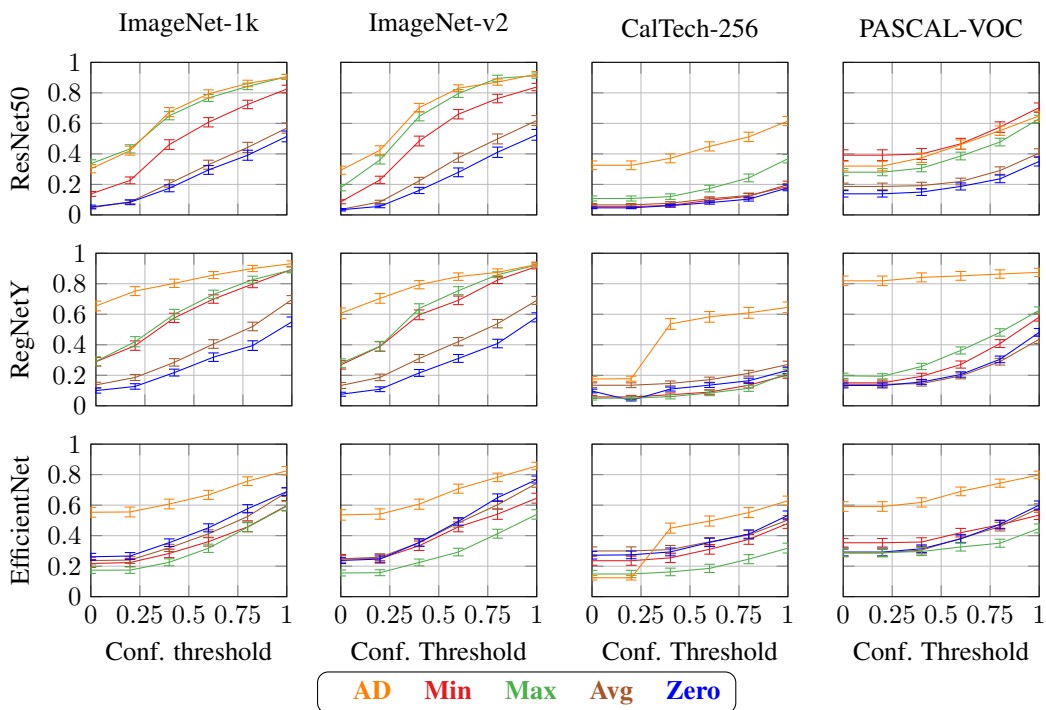

Figure 5: $\rho$-robustness (rows, from $0$ to $1$ in steps of $0.2$) of explanation against non-informative background on ImageNet-1k, ImageNet-1k_v2, CalTech(CT)-256 and PASCAL-VOC for our models, with different confidence thresholds $\gamma$. AD explanations are consistently more robust than ones computed using masking values, for all masking values and with all confidence thresholds. Error bars represents Standard Error of the Mean (SEM).

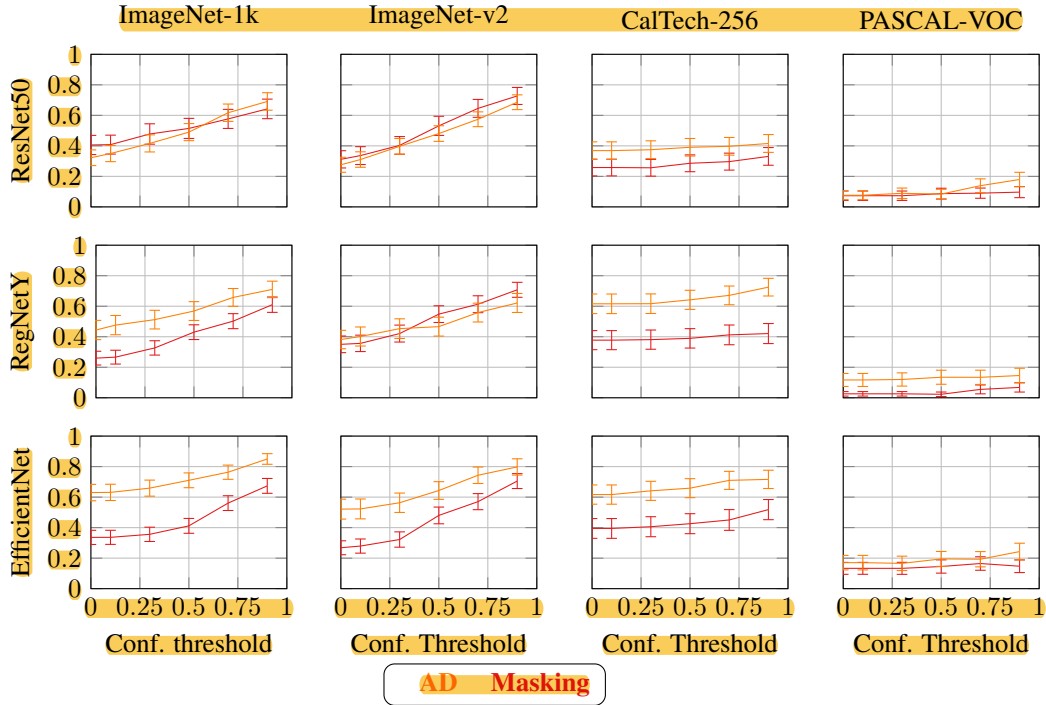

Figure 6: $\rho$-robustness (rows, from $0$ to $1$ in steps of $0.2$) calculated via LIME. Note that the AD explanations are still better than the marking strategy employed by LIME, the decreased robustness can be attributed to the crudeness of the saliency calculation of LIME compared to REX. Error bars represent Standard Error of the Mean (SEM).

Table 1: Average size difference of CONVAD explanation vs best-performing masking value

| Model (Dataset) | Thresholds (%) | | | | | |
|---|---|---|---|---|---|---|
| | 0.9 | 0.7 | 0.5 | 0.3 | 0.1 | 0.0 |
| ResNet-50 (ImageNet-1k) | 12.80 | 9.89 | 2.52 | 0.70 | 4.06 | 4.20 |
| ResNet-50 (IN-1k_v2) | 15.27 | 11.28 | 1.65 | 1.31 | 4.32 | 3.95 |
| ResNet-50 (Caltech-256) | 11.04 | 10.03 | 4.98 | 4.48 | 7.42 | 7.42 |
| ResNet-50 (PASCAL-VOC) | 2.43 | 1.41 | 1.46 | 0.96 | 0.09 | −5.79 |
| RegNetY-12GF (ImageNet-1k) | 6.23 | 6.10 | 5.38 | 4.93 | 6.54 | 4.37 |
| RegNetY-12GF (IN-1k_v2) | 1.98 | 1.75 | 2.40 | 2.26 | 1.92 | 1.86 |
| RegNetY-12GF (CalTech-256) | 19.53 | 18.00 | 16.20 | 15.14 | 15.78 | 13.76 |
| RegNetY-12GF (PASCAL-VOC) | 14.58 | 13.09 | 12.33 | 11.69 | 11.61 | 11.40 |
| EfficientNet-V2-S (ImageNet-1k) | 7.83 | 6.42 | 6.71 | 6.41 | 6.03 | 6.98 |
| EfficientNet-V2-S (IN-1k_v2) | 8.60 | 4.92 | 4.85 | 4.24 | 4.44 | 4.84 |
| EfficientNet-V2-S (CalTech-256) | 7.06 | 6.02 | 4.59 | 4.49 | 11.77 | 12.69 |
| EfficientNet-V2-S (PASCAL-VOC) | 6.96 | 6.73 | 5.23 | 5.31 | 5.00 | 5.55 |

contrast, AD consistently is the best or near-best performing method across all datasets and models, and does not require any decision wrt the best occlusion value.

## 5.2 ROBUSTNESS AGAINST IID BACKGROUNDS

A similar robustness pattern is found when placing inserting explanations over IID backgrounds (see the plots in Appendix A). However, the overall level of robustness is much lower, with AD having approximately 20% robustness at $\gamma = 0.9$. This behavior is relatively easy to explain. Every model tested *must* provide a classification. Even a solid color background has a classification, albeit of low confidence. Examination of the output tensor for solid backgrounds (see Section 5.1) reveals a close-to-flat probability distribution, making it relatively easy to change the top class by injecting an explanation for a different class. This distribution flatness does not hold for IID images: some classes have a higher probability than all the others. It therefore requires more "work" from the injected explanation in order to overturn the background's classification. Given the relatively small size of causal explanations in general, the fact that AD manages to overwrite the classifications of as many as 20% of the IID backgrounds suggests a higher meaningfulness of AD explanations.

## 5.3 AVERAGE SIZE OF EXPLANATIONS

Table 1 provides a summary of the results and insights into the characteristics of CONVAD explanations (see the appendix for the detailed breakdown). CONVAD explanations are generally somewhat *larger*, though the size varies across models and datasets. The smallest increase in size is 0.9% ($\gamma = 0.1$, ResNet-50 PASCAL-VOC) and the largest increase is 19.53% ($\gamma = 0.9$, RegNetY-12GF CalTech-256). The results observed at lower confidence levels are more mixed, which is likely to be due to the low robustness of these explanations.

The results demonstrate that CONVAD explanations utilize a larger portion of the input in order to provide an explanation of certain confidence. This observation suggests a solution to the problem highlighted in Kelly et al. (2025): models, especially large models, often use only a tiny fraction of the input for a classification, resulting in extremely small explanations. While we have not performed a user study, and it is out of scope for this paper, larger and more robust explanations generated by CONVAD are likely to increase trust in both an explanation and a model.

## 5.4 CONFIDENCE OF EXPLANATIONS VS THE CLASSIFICATION

The full confidence results for AD explanations vs occlusion-based ones with different occlusion values are presented in the appendix (Appendix G). Overall, the average confidences in explanation are similar between the different methods (more so for ImageNet-1k and ImageNet-1k_v2 and somewhat less for CalTech-256 and PASCAL-VOC). An interesting observation is that AD explanations

are the closest to the required confidence thresholds w.r.t the confidence in the overall classification, while also being more robust.

## 6 RELATED WORK

**Explainability for AI models** There is a plethora of post-hoc explainability methods for DNNs, both white-box (model-specific) ones (Selvaraju et al., 2017; Chattopadhay et al., 2018; Bach et al., 2015; Shrikumar et al., 2017; Sundararajan et al., 2017; Springenberg et al., 2015; Simonyan et al., 2013), and black-box (model-agnostic) (Chockler et al., 2024; Ribeiro et al., 2016; Lundberg & Lee, 2017; Petsiuk et al., 2018; Ribeiro et al., 2018; Zeiler & Fergus, 2014). These methods calculate saliency landscapes, summarizing the contribution of the different input features towards the model's decision. White-box or model-specific methods calculate this attribution by analyzing the intermediate layers and the weights of activation maps across different layers. Black-box methods compute attribution using repeated calls to the model over perturbed variants of the inputs.

**Alternative perturbation strategies** Various methods have been propose to create mutants that more closely reflect in-distribution samples. Generative in-painting is touted as a popular method for the same, specially in high-dimensional settings such as images and complex signals. Previous iterations of such perturbations utilized GANs and VAEs (Shih et al. (2021); Xiang et al. (2023)), however, newer methods rely on Diffusion models (Ademi et al. (2025); Solís-Martín et al. (2025); Jeanneret et al. (2022)). Generative methods, however tend to incorrectly attribute importance to correlated features with no predictive strength as well as are computational expensive to train. maintain and run. Optimization-based perturbations are another ubiquitous set of methods, where the goal is to learn the optimal mask for a given instance. Meaningful perturbations (Fong & Vedaldi (2017)) is a seminal work in this domain, where the problem is formulated to find the smallest mask $m$ that causes the score $f_c(\Phi(x_0; m))$ to drop significantly. Such methods run the risk of triggering artifacts in the model, which leads to nonsensical or unexpected outputs.

**Metrics for attribution-based XAI methods** Numerous metrics for evaluating XAI exist that measure properties that a good explanation should posses, faithfulness (adherence to true decision-making process) (Zheng et al. (2024); Hooker et al. (2019); Yeh et al. (2019)), robustness(stability against irrelevant noise) (Vascotto et al. (2024); Agarwal et al. (2022))and plausibility/interpretability(Wang et al. (2020); Bohle et al. (2021); Jacovi & Goldberg (2020)). The concept of faithfulness is poorly defined as preexisting definitions of the same do not define its meaning concretely with respect to the model, and therefore it brings forth a degree of ambiguity. Furthermore, as Jacovi & Goldberg (2020); Bhusal et al. (2025) state, using interpretability is not necessarily a good metric for evaluating explanations as the decision making process of model can differ from human intuition and an explanation which uncovers the same is not a bad explanation. Therefore, we are interested primarily in evaluating the robustness of the explanations and subsequently introduce $\rho$-robustness, which contextualizes the idea of robustness within the framework of actual causality by calculating in what ratio of contexts does the minimal explanation still hold as a valid explanation. $\rho$-robustness goes one step further than previous notions of robustness as it can be evaluated against any possible context, whereas previous notions look at adjacent examples.

## 7 CONCLUSIONS

We have introduced AD, a forward-pass paradigm which obviates the need for occlusions in post-hoc explainability. We proved theoretical results regarding the explanations calculated in AD. We have presented CONVAD, a drop-in mechanism that can be added to any trained CNN and which, without any further training or fine-tuning, provides more robust explanations that also adhere more strictly to the confidence threshold of the model.

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

# Appendix

In this appendix to our work on Activation Deactivation, we provide:

# A ROBUSTNESS AGAINST IID EXAMPLES

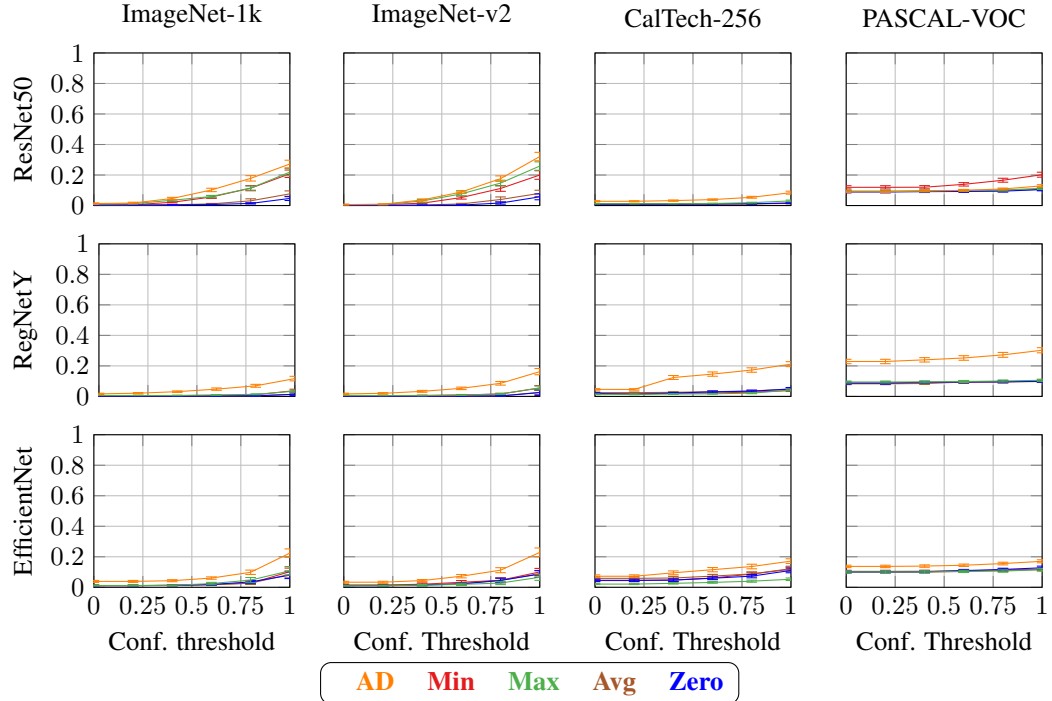

Figure A1: $\rho$-robustness (rows) of explanation against IID background on ImageNet-1k, ImageNet-1k_v2 and CalTech-256 for ResNet-50, RegNet-Y and EfficientNet_v2 models. Class robustness of the AD explanation against the different masking values is more consistent and more robust, especially for RegNetY and EfficientNet-v2. Error bars represents Standard Error of the Mean (SEM).

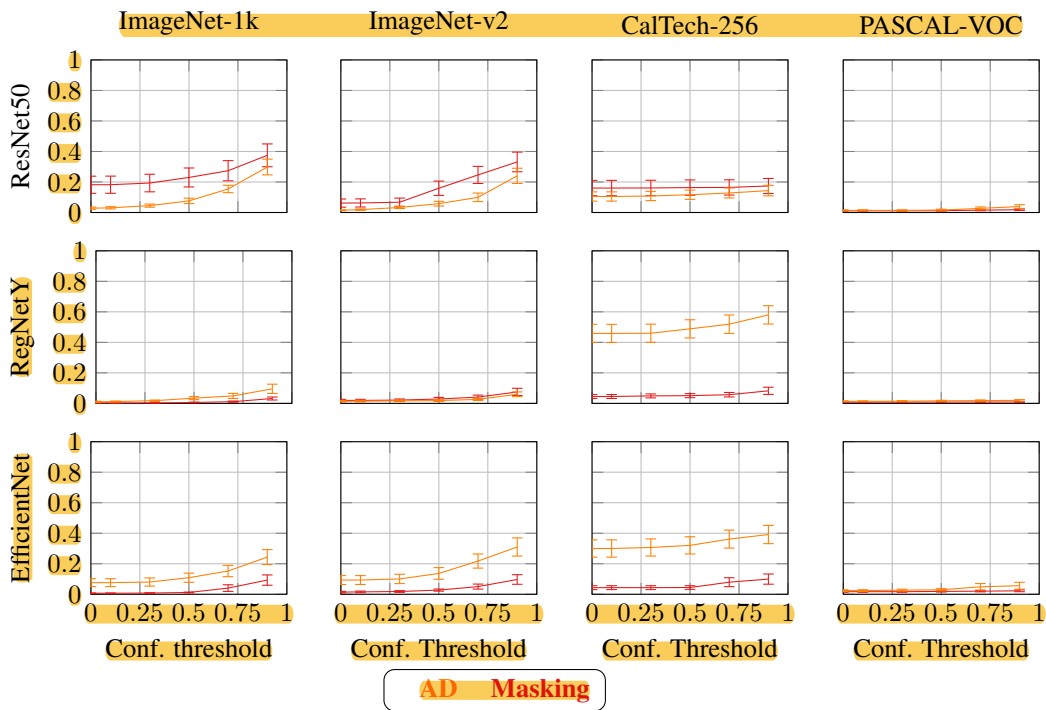

Figure A2: $\rho$-robustness (rows) of explanation against IID background on ImageNet-1k, ImageNet-1k_v2 and CalTech-256 for ResNet-50, RegNet-Y and EfficientNet_v2 models calculated via LIME. Error bars represents Standard Error of the Mean (SEM).

## B  BACKGROUND ON ACTUAL CAUSALITY

In what follows, we briefly introduce the relevant definitions from the theory of actual causality. The reader is referred to Halpern (2019) for further reading. We assume that the world is described in terms of variables and their values. Some variables may have a causal influence on others. This influence is modeled by a set of *structural equations*. It is conceptually useful to split the variables into two sets: the *exogenous* variables $\mathcal{U}$, whose values are determined by factors outside the model, and the *endogenous* variables $\mathcal{V}$, whose values are ultimately determined by the exogenous variables. The structural equations $\mathcal{F}$ describe how these values are determined. A *causal model* $M$ is described by its variables and the structural equations. We restrict the discussion to acyclic (recursive) causal models. A *context* $\vec{u}$ is a setting for the exogenous variables $\mathcal{U}$, which then determines the values of all other variables. We call a pair $(M, \vec{u})$ consisting of a causal model $M$ and a context $\vec{u}$ a *(causal) setting*. An intervention is defined as setting the value of some variable $X$ to $x$, and essentially amounts to replacing the equation for $X$ in $\mathcal{F}$ by $X = x$. A *probabilistic causal model* is a pair $(M, \Pr)$, where $\Pr$ is a probability distribution on a given set of contexts $\mathcal{K}$.

A causal formula $\psi$ is true or false in a setting. We write $(M, \vec{u}) \models \psi$ if the causal formula $\psi$ is true in the setting $(M, \vec{u})$. Finally, $(M, \vec{u}) \models [\vec{Y} \leftarrow \vec{y}]\varphi$ if $(M_{\vec{Y}=\vec{y}}, \vec{u}) \models \varphi$, where $M_{\vec{Y}\leftarrow\vec{y}}$ is the causal model that is identical to $M$, except that the variables in $\vec{Y}$ are set to $Y = y$ for each $Y \in \vec{Y}$ and its corresponding value $y \in \vec{y}$.

A standard use of causal models is to define *actual causation*: that is, what it means for some particular event that occurred to cause another particular event. There have been a number of definitions of actual causation given for acyclic models (*e.g.*, Beckers (2021); Glymour & Wimberly (2007); Hall (2007); Halpern & Pearl (2005); Halpern (2019); Hitchcock (2001; 2007); Weslake (2015); Woodward (2003)). In this paper, we focus on the definitions in Halpern (2019), which we briefly review below. The events that can be causes are arbitrary conjunctions of primitive events.

**Definition 8** (Actual cause). $\vec{X} = \vec{x}$ is an *actual cause* of $\varphi$ in $(M, \vec{u})$ if the following three conditions hold:

AC1.  $(M, \vec{u}) \models (\vec{X} = \vec{x})$ and $(M, \vec{u}) \models \varphi$.

AC2.  There is a a setting $\vec{x}'$ of the variables in $\vec{X}$, a (possibly empty) set $\vec{W}$ of variables in $\mathcal{V} - \vec{X}'$, and a setting $\vec{w}$ of the variables in $\vec{W}$ such that $(M, \vec{u}) \models \vec{W} = \vec{w}$ and $(M, \vec{u}) \models [\vec{X} \leftarrow \vec{x}', \vec{W} \leftarrow \vec{w}]\neg\varphi$, and moreover

AC3.  $\vec{X}$ is minimal; there is no strict subset $\vec{X}'$ of $\vec{X}$ such that $\vec{X}' = \vec{x}''$ can replace $\vec{X} = \vec{x}'$ in AC2, where $\vec{x}''$ is the restriction of $\vec{x}'$ to the variables in $\vec{X}'$.

In the special case that $\vec{W} = \emptyset$, we get the but-for definition. A variable $X$ in an actual cause $\vec{X}$ is called a *part of a cause*. In what follows, following Halpern we state that *part of a cause is a cause*. The *degree of responsibility* quantifies actual causality Chockler & Halpern (2004); Halpern (2019). Specifically, the degree of responsibility of a part of a cause $X = x$ in $\varphi$ is defined as $1/(|\vec{X}| + |\vec{W}|)$, where $X \in \vec{X}$, $X$ is assigned $x$ in $\vec{x}$, and $\vec{X} = \vec{x}$ and $\vec{W}$ satisfy Definition 8.

In this paper, we follow the construction of depth 2 causal models with independent input variables, introduced in Chockler & Halpern (2024) for black-box explainability of image classifiers. Essentially, we view the neural network as a probabilistic causal model with a single internal node, the set of endogenous variables being the set of the features of the input, and a single output node $O$. The (unknown) equation for $O$ determines the output of the neural network as a function of the input. Thus, the causal network has depth 2, with the exogenous variables determining the feature variables, and the feature variables determining the output variable. We assume *causal independence* between the feature variables $\vec{V}$; this is true in image classification and is a reasonable approximation in other domains we consider in this paper. We also note that feature independence is a common assumption in explainability tools.

It is easy to see that in this setting, in the condition AC2 of Definition 8, the set $\vec{W}$ is empty. That is, we are looking for a setting $\vec{x}'$ of the variables in $\vec{X}$ such that $(M, \vec{u}) \models [\vec{X} \leftarrow \vec{x}']\neg\varphi$. Note also that the degree of responsibility of $X = x$ in these models is $1/|\vec{X}|$.

## C    PROOF OF THEOREM 1

**Assumption 1.** The dimensionality reductions/expansions are due to parametric operations or external additive/subtractive procedures to the intermediate representations.

For Theorem 1, there are two cases that we need to account for, one of them being the case where the intermediate representations, $\mathbf{z}$, changes in shape between successive convolutions in the model. The above assumption is used to prove the results for the broadest set of convolutional architectures, as parameterized alteration of the shape of $\mathbf{z}$, is still dependent on the input features, and thus, is subject to AD. This setting covers most/all of the possible settings in which the shape of the intermediate representations, such as downsampling methods (max pooling, average pooling, global pooling etc.) or other feature-dependent reductions, upsampling methods or other feature-dependent expansions (such as replication of representations, interpolation) and external additions not informed by the representations (i.e. padding).

**Equivalence to regular network in the unmasked case.**    Given a CNN $\mathcal{N}$, we denote by $\mathcal{N}'$ the result of application of CONVAD on $\mathcal{N}$. Then, $\mathcal{N}'$ performs identically to $\mathcal{N}$ for inputs without occlusions.

**Theorem 1.** *Given a model $\mathcal{N}$ and the result $\mathcal{N}'$ of applying CONVAD to $\mathcal{N}$, the output of $\mathcal{N}'$ is equal to the output of $\mathcal{N}$ on all inputs without occlusions (that is, without an accompanying binary mask).*

*Proof.* Passing only an input image, $\mathbf{x}$, is equivalent to passing the tuple $(\mathbf{x}, J_{m \times n})$, where $J_{m \times n}$ is a matrix of all ones and $\mathbf{x}, J_{m \times n} \in \mathbb{R}^{m \times n}$.

The checkpoints in the CONVAD model, $\mathcal{N}'$, occur at least once between each two successive convolutional layers. We can separate the checkpoints to two cases:

**Case 1:**    $shape(\mathcal{N}_{(i)}(x)) = shape(\mathcal{N}_{(j-1)}(x))$ between successive convolutional layers, $\mathcal{N}_i$ and $\mathcal{N}_j$.

This case holds for the set of parametric layers $\{\mathcal{N}_{i+1}...\mathcal{N}_{j-1}\}$ that do not alter the shape of the representations. Example of such operations include activation functions (ReLU, Tanh, GeLU etc.), normalization layers (Batch or Layer Normalization), regularization layers (Dropout) etc. Note, this case also holds if $\mathcal{N}_j$ is an immediate successor to $\mathcal{N}_i$ (i.e. $\mathcal{N}_i = \mathcal{N}_{j-1}$), which occurs in cases where we perform successive convolutions.

Let $\mathcal{N}_{i..j-1} \in \mathbb{R}^{C \times K_H \times K_w}$, according to the AD definition, we require a position-attribution function at the checkpoint. For CONVAD, $\Phi_{j-1}(\cdot) \in \mathbb{R}^{1 \times d}$, it is defined as an accompanying mean-value convolution filter which we define at the checkpoint after layer $j - 1$.

The value of each element in $\Phi_{\mathcal{N}_{j-1}}$ is set to $\frac{1}{K_H \times K_W}$

We attain the layer output, $O^{(i)}$, by convolving it by with convolution filter $\mathcal{N}^{(i)}$

$$O_n^{(i)}[k] = (\mathbf{z} * \mathcal{N}_n^{(i)})[k] = \sum_{i=0}^{K_H-1} \sum_{j=0}^{K_W-1} \mathbf{z}(h + i, w + j) \cdot \mathcal{N}_n^{(i)}(h, w)$$

Where $O_n^{(i)}[k]$ is the $k^{th}$ entry of the $n^{th}$ channel of the output of the convolutional layer.

After the convolution, all intermediate operations prior to the next convolution is performed wherein the dimensionality of the output is retained. Let $f : \mathbb{R}^d \to \mathbb{R}^d$ be the composition of these intermediate operations and therefore, we obtain the output till layer $j - 1$:

$$O^{(j-1)} = f(O^{(i)})$$

In parallel, we can calculate the output of the position attribution function for $\mathcal{N}'$, which in our case is the convolution of $J_{m \times n}$ with the filter $\Phi_{j-1}$.

$$\Phi_{j-1}(k, M) = (J_{m \times n} * \Phi_{j-1})[k] = \sum_{i=0}^{K_H-1} \sum_{j=0}^{K_W-1} J_{m \times n}(h+i, w+j) \cdot \Phi_{j-1}(h, w)$$

$$= K_H \times K_W \cdot \left(1 \cdot \frac{1}{K_H \times K_W}\right)$$

$$= 1$$

Thus we obtain $\Phi_{j-1}(M)$ such that all entries are 1 and $dim(\Phi_{j-1}(M)) = dim(O_n^{(j-1)})$. We then get the updated output by performing a Hadamard product between $O^{(j-1)}$ and $\Phi_{j-1}(M)$, which can be done by vectorizing $\Phi_{j-1}(M)$.

$$\Phi_{j-1}^{\oplus}(M) \in \mathbb{R}^{dim(O^{(j-1)})}$$

$$O'^{(j-1)} = O^{(j-1)} \odot \Phi_{j-1}^{\oplus}(M) = O^{(j-1)} \odot J_{dim(O^{(j-1)})} = O^{(j-1)}$$

Which is equivalent to the output from the network $\mathcal{N}$ prior to the convolutional layer $\mathcal{N}_{(j)}$. It is clear that this is repeated at each checkpoint in $\mathcal{N}'$.

**Case 2:** $shape(\mathcal{N}_i(x)) \neq shape(\mathcal{N}_{j-1}(x))$ between successive convolutional layers, $\mathcal{N}_i$ and $\mathcal{N}_j$.

This case occurs either when the set of parametric intermediate operations $\{\mathcal{N}_{i+1}...\mathcal{N}_{j-1}\}$ alter the dimension of the intermediate representation $\mathbf{z}$ by reducing or expanding it, or via an independent external additive or subtractive process. This covers most, if not all possible settings where the shape of the representation is modified. Checkpoints are defined at each of the dimensionality altering positions.

We first examine the case of checkpoints at parametric operations (i.e. downsampling methods such as pooling and upsampling methods such as interpolation).

For such checkpoints, the tuple $(O^{(j-1)}, J_D), O^{(j-1)} \in \mathbb{R}^{N \times D}, J_D \in \mathbb{R}^{1 \times D}$ is the input representation and its corresponding mask before the dimension altering functions, this leads to:

$$\mathcal{N}_{(j)}(O^{(j-1)}) = O^{(j)} \in \mathbb{R}^{N \times D'}$$

Similar to the position attribution calculation in case 1,

$$\Phi_j(J_D) = J_{D'} \in \mathbb{R}^{1 \times D'}$$

Therefore, similar to case 1, the output for the CONVAD model $\mathcal{N}'^{(j)}$ is:

$$O'^{(j)} = O^{(j)} \odot \Phi_{j-1}^{\oplus}(M) = O^{(j)} \odot \Phi_j^{\oplus}(J_{D'}) = O^{(j)}$$

Which is equivalent to the output of the network $\mathcal{N}$.

For the case of external additive effects, these are considered to be independent features and thus can be represented as an input. It can similarly be represented as a 2-tuple:

$$(\mathbf{z}'^{(j)}, J_{D'}), \mathbf{z}'^{(j)} \in \mathbb{R}^{N \times D'}, J_{D'} \in \mathbb{R}^{1 \times D'}$$

Where $\mathbf{z}'^{(j)}$ is the set of representations that are going to be added to $O^{(j-1)}$. Given that the representations to be added are independent of the model's intermediate features, masking or unmasking its effects is an arbitrary decision. For the unmasked scenario, $J^{D'}$ is the accompanying mask for the external representations. We then apply the same combination function over representations and the binary masks.

$$\mathbf{z}'' = \text{combine}(O^{(j-1)}, \mathbf{z}'), \mathbf{z}'' \in \mathbb{R}^{N \times D''}$$

$$M = J_{D''} = \text{combine}(J_D, J_{D'}), J_{D''} \in \mathbb{R}^{1 \times D''}$$

For external subtractive procedures, the scenario is nearly identical, instead of the combine function, we have a reduce function.

$$\mathbf{z}'' = \text{reduce}(O^{(j-1)}), \mathbf{z}'' \in \mathbb{R}^{N \times D''}$$

$$M = J_{D''} = \text{reduce}(J_D, ), J_{D''} \in \mathbb{R}^{1 \times D''}$$

The output of the regular network, $\mathcal{N}$ on the updated representations would be:

$$\mathcal{N}_{(j)}(\mathbf{z}'') = O^{(j)}$$

The output of the AD network, $\mathcal{N}'$, is:

$$O'^{(j)} = O^{(j)} \odot \Phi_j^{\oplus}(M) = O^{(j)} \odot \Phi_j^{\oplus}(J_{D''}) = O^{(j)}$$

Which is equivalent to the output of the network $\mathcal{N}$.

$\square$

## D   PROOF OF COROLLARY 1

We first state the definition of a prime-implicant or abductive explanation:

**Definition 9** (Prime Implicant/Abductive explanation (Marques-Silva et al. (2021))). An abductive, or prime-implicant (PI), explanation is a subset-minimal set of features $\mathcal{X} \subseteq \mathcal{F}$, which, if assigned the values $v$ dictated by the instance $(v, c)$ are sufficient for the prediction $c$.

$$\forall (x \in \mathbb{F})[\bigwedge_{i \in \mathcal{X}} (x_i = v_i)] \rightarrow (\mathcal{N}(x) = c)$$

We now restate Corollary 1:

**Corollary 1.** *Causal explanations over the set of all possible contexts are prime-implicant explanations, we have that* AD-*explanations are also prime-implicant explanations (but the inverse does not necessarily hold).*

*Proof.* For an instance $(v, c)$, a set of variables $\mathcal{X} \subseteq \mathcal{F}$ and their values as defined by $v$ satisfy Definition 9 iff they satisfy EXIC1 in Definition 1. Moreover, $\mathcal{X}$ is subset-minimal in $\mathcal{F}$ iff it satisfies EXIC2 in Definition 1. Note that EXIC3 does not have an equivalent clause in the definition of abductive explanations, hence the other direction does not necessarily hold. $\square$

## E   MODEL ACCURACY

Table E1: Model accuracy

| Model | Accuracy on sample set% ↑ | | | |
|---|---|---|---|---|
| | IN-1k | IN-1k v2 | CalTech-256 | PASCAL-VOC |
| ResNet-50 | 80.0 | 84.0 | 85.3 | 83.68 |
| RegNetY-12GF | 86.7 | 86.0 | 82.0 | 84.51 |
| EfficientNet-V2-S | 86.0 | 86.0 | 85.3 | 82.61 |

# F  AVERAGE SIZE OF EXPLANATIONS

Table F1: Average explanation size for ResNet on ImageNet-1k

| Masking Value | Threshold | | | | | |
|---|---|---|---|---|---|---|
| | 0.9 | 0.7 | 0.5 | 0.3 | 0.1 | 0.0 |
| Min | 21.84% | 16.12% | 8.18% | 6.11% | 7.92% | 5.76% |
| Max | 10.97% | 8.46% | 12.60% | 10.56% | 3.95% | 2.91% |
| Zero | 20.27% | 15.00% | 6.86% | 5.31% | 6.45% | 6.97% |
| Avg | 13.04% | 10.14% | 11.79% | 9.04% | 4.31% | 3.54% |
| AD | 23.78% | 18.34% | 15.12% | 11.26% | 8.02% | 7.11% |

Table F2: Average explanation size for ResNet on IN_V2

| Masking Value | Threshold | | | | | |
|---|---|---|---|---|---|---|
| | 0.9 | 0.7 | 0.5 | 0.3 | 0.1 | 0.0 |
| Min | 22.64% | 17.24% | 7.42% | 5.67% | 6.76% | 4.95% |
| Max | 11.23% | 7.88% | 13.19% | 9.81% | 3.13% | 2.25% |
| Zero | 22.09% | 16.66% | 6.17% | 4.82% | 5.91% | 5.73% |
| Avg | 13.54% | 10.06% | 12.73% | 9.08% | 3.69% | 2.87% |
| AD | 26.50% | 19.15% | 14.84% | 11.12% | 7.45% | 6.20% |

Table F3: Average explanation size for ResNet on CT-256

| Masking Value | Threshold | | | | | |
|---|---|---|---|---|---|---|
| | 0.9 | 0.7 | 0.5 | 0.3 | 0.1 | 0.0 |
| Min | 12.76% | 10.73% | 5.80% | 5.19% | 8.12% | 5.24% |
| Max | 8.30% | 6.33% | 9.30% | 8.50% | 4.53% | 4.53% |
| Zero | 8.77% | 7.29% | 5.61% | 4.89% | 5.24% | 8.12% |
| Avg | 8.08% | 6.67% | 6.64% | 5.76% | 4.88% | 4.88% |
| AD | 19.34% | 16.35% | 14.28% | 12.98% | 11.95% | 11.95% |

Table F4: Average explanation size for ResNet on PASCAL-VOC

| Masking Value | Threshold | | | | | |
|---|---|---|---|---|---|---|
| | 0.9 | 0.7 | 0.5 | 0.3 | 0.1 | 0.0 |
| Min | 5.92% | 3.93% | 2.56% | 2.28% | 2.66% | 8.54% |
| Max | 4.60% | 2.68% | 3.31% | 2.84% | 1.60% | 1.60% |
| Zero | 13.26% | 10.45% | 2.17% | 1.75% | 8.54% | 2.66% |
| Avg | 4.53% | 3.11% | 9.01% | 8.60% | 2.17% | 2.17% |
| AD | 8.35% | 5.34% | 4.02% | 3.23% | 2.75% | 2.75% |

Table F5: Average explanation size for RegNet on ImageNet-1k

| Masking Value | Threshold | | | | | |
|---|---|---|---|---|---|---|
| | 0.9 | 0.7 | 0.5 | 0.3 | 0.1 | 0.0 |
| Min | 10.16% | 7.16% | 5.68% | 2.90% | 4.17% | 1.83% |
| Max | 9.68% | 6.85% | 5.89% | 4.97% | 2.23% | 3.33% |
| Zero | 6.70% | 4.78% | 3.47% | 3.13% | 3.93% | 2.17% |
| Avg | 6.29% | 4.33% | 3.92% | 4.70% | 2.54% | 3.66% |
| AD | 15.92% | 12.95% | 11.27% | 9.89% | 8.76% | 7.69% |

Table F6: Average explanation size for RegNet on IN-1k_V2

| Masking Value | Threshold | | | | | |
|---|---|---|---|---|---|---|
| | 0.9 | 0.7 | 0.5 | 0.3 | 0.1 | 0.0 |
| Min | 3.69% | 1.73% | 1.94% | 1.80% | 0.79% | 1.20% |
| Max | 3.67% | 2.58% | 1.38% | 1.01% | 0.89% | 0.68% |
| Zero | 2.39% | 1.54% | 1.27% | 1.13% | 1.45% | 0.77% |
| Avg | 2.56% | 2.44% | 2.15% | 1.63% | 1.35% | 1.27% |
| AD | 5.65% | 4.33% | 3.78% | 3.27% | 2.82% | 2.55% |

Table F7: Average explanation size for RegNet on CT-256

| Masking Value | Threshold | | | | | |
|---|---|---|---|---|---|---|
| | 0.9 | 0.7 | 0.5 | 0.3 | 0.1 | 0.0 |
| Min | 7.39% | 6.45% | 5.34% | 3.90% | 5.28% | 3.75% |
| Max | 7.87% | 6.25% | 5.66% | 5.40% | 3.75% | 4.00% |
| Zero | 6.00% | 5.36% | 4.27% | 3.79% | 4.00% | 3.55% |
| Avg | 6.63% | 5.17% | 4.69% | 4.24% | 3.55% | 5.28% |
| AD | 26.16% | 23.18% | 20.88% | 19.38% | 19.33% | 19.04% |

Table F8: Average explanation size for RegNet on PASCAL-VOC

| Masking Value | Threshold | | | | | |
|---|---|---|---|---|---|---|
| | 0.9 | 0.7 | 0.5 | 0.3 | 0.1 | 0.0 |
| Min | 3.07% | 2.18% | 1.84% | 1.22% | 1.00% | 1.09% |
| Max | 3.35% | 2.52% | 1.59% | 1.15% | 1.09% | 1.30% |
| Zero | 2.52% | 1.89% | 1.48% | 1.21% | 1.30% | 1.16% |
| Avg | 2.86% | 1.81% | 1.54% | 1.51% | 1.16% | 1.00% |
| AD | 17.93% | 15.61% | 13.92% | 12.84% | 12.70% | 12.70% |

Table F9: Average explanation size for EFN-V2 on ImageNet-1k

| Masking Value | Threshold | | | | | |
|---|---|---|---|---|---|---|
| | 0.9 | 0.7 | 0.5 | 0.3 | 0.1 | 0.0 |
| Min | 18.17% | 12.93% | 11.35% | 10.51% | 9.75% | 9.31% |
| Max | 17.27% | 13.82% | 11.67% | 10.80% | 9.50% | 10.25% |
| Zero | 18.41% | 14.08% | 11.85% | 10.49% | 10.25% | 9.28% |
| Avg | 18.06% | 13.34% | 12.46% | 11.28% | 9.38% | 9.66% |
| AD | 26.23% | 20.51% | 18.56% | 16.90% | 16.28% | 16.26% |

Table F10: Average explanation size for EFN-V2 on IN-1k_V2

| Masking Value | Threshold | | | | | |
|---|---|---|---|---|---|---|
| | 0.9 | 0.7 | 0.5 | 0.3 | 0.1 | 0.0 |
| Min | 16.35% | 13.28% | 11.36% | 10.13% | 9.12% | 8.97% |
| Max | 16.98% | 13.66% | 12.28% | 10.08% | 9.80% | 9.36% |
| Zero | 15.09% | 13.70% | 11.67% | 10.72% | 9.49% | 9.05% |
| Avg | 15.99% | 12.35% | 10.93% | 10.11% | 9.14% | 9.73% |
| AD | 23.69% | 18.62% | 16.52% | 14.97% | 13.93% | 13.89% |

Table F11: Average explanation size for EFN-V2 on CT-256

| Masking Value | Threshold | | | | | |
|---|---|---|---|---|---|---|
| | 0.9 | 0.7 | 0.5 | 0.3 | 0.1 | 0.0 |
| Min | 17.06% | 15.22% | 13.76% | 11.60% | 11.62% | 11.27% |
| Max | 17.90% | 15.51% | 14.04% | 12.55% | 11.62% | 12.89% |
| Zero | 15.63% | 14.15% | 13.66% | 12.15% | 12.36% | 11.44% |
| Avg | 18.57% | 14.79% | 13.21% | 12.03% | 11.47% | 11.62% |
| AD | 22.69% | 20.17% | 18.25% | 16.64% | 24.12% | 24.12% |

Table F12: Average explanation size for EFN-V2 on PASCAL-VOC

| Masking Value | Threshold | | | | | |
|---|---|---|---|---|---|---|
| | 0.9 | 0.7 | 0.5 | 0.3 | 0.1 | 0.0 |
| Min | 7.35% | 5.91% | 3.48% | 2.70% | 4.30% | 2.81% |
| Max | 6.71% | 5.22% | 5.29% | 4.57% | 2.81% | 3.09% |
| Zero | 5.74% | 4.25% | 3.94% | 3.05% | 3.09% | 2.54% |
| Avg | 5.81% | 4.32% | 3.65% | 3.24% | 2.54% | 4.30% |
| AD | 12.70% | 10.98% | 9.17% | 8.36% | 8.09% | 8.09% |

# G CONFIDENCE SCORES OF EXPLANATIONS AND CONFIDENCE DIFFERENCE ACROSS DIFFERENT BACKGROUNDS

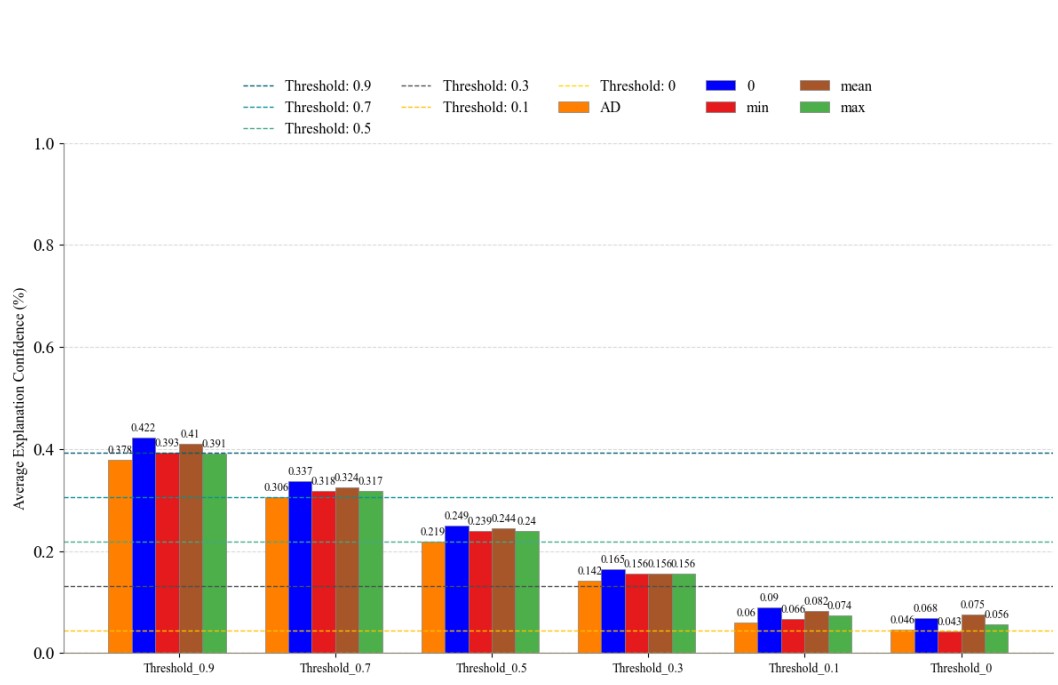

Figure G1: Average Explanation Confidence of different methods for ResNet on ImageNet-1k

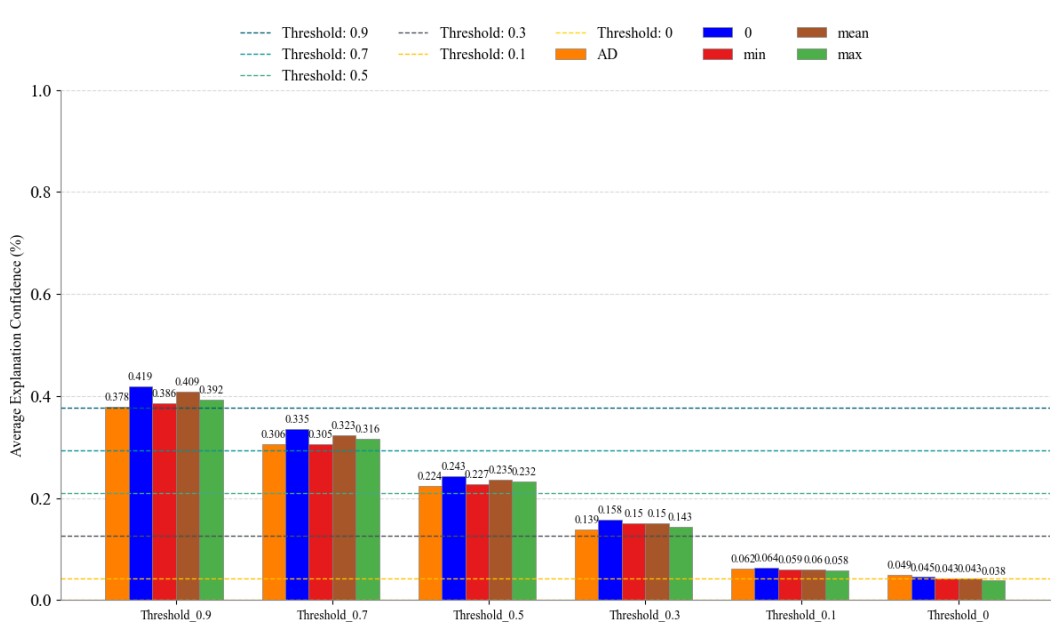

Figure G2: Average Explanation Confidence of different methods for ResNet on IN-1k_V2

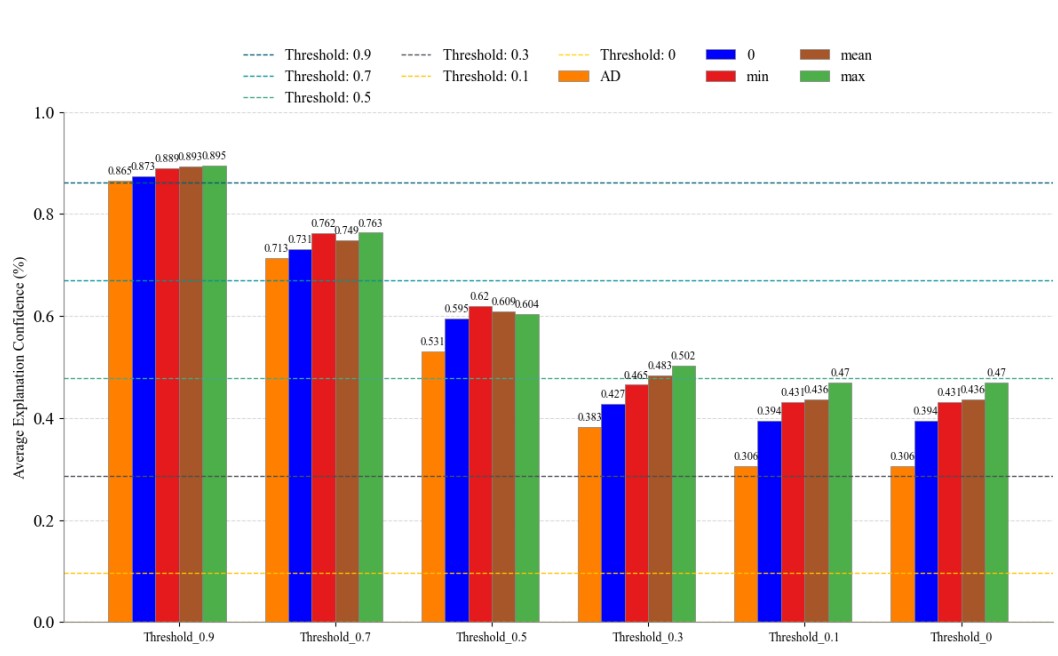

Figure G3: Average Explanation Confidence of different methods for ResNet on CalTech-256

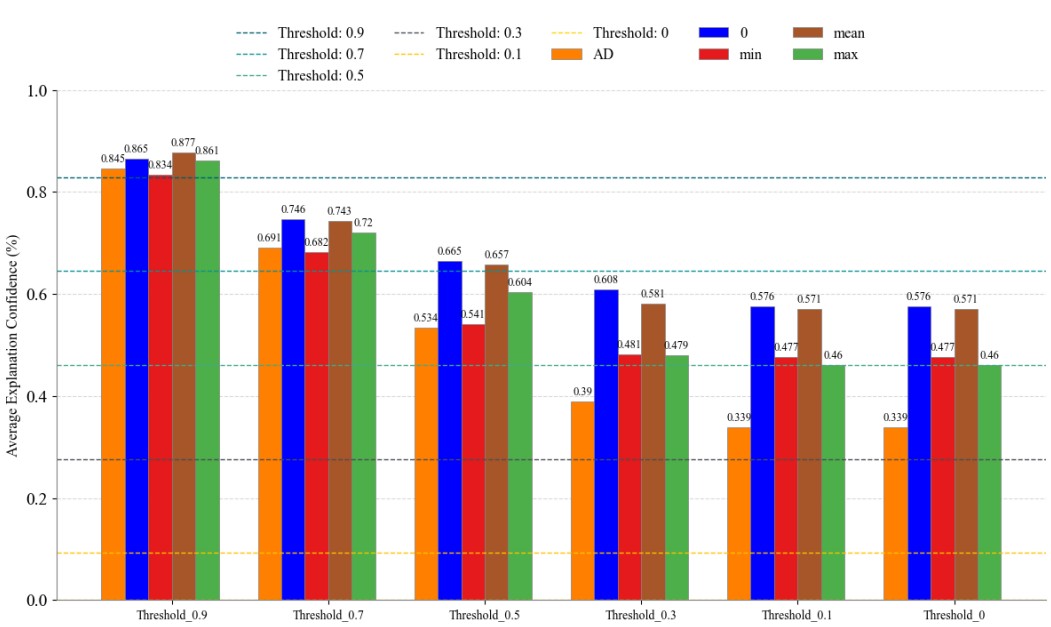

Figure G4: Average Explanation Confidence of different methods for ResNet on PASCAL-VOC

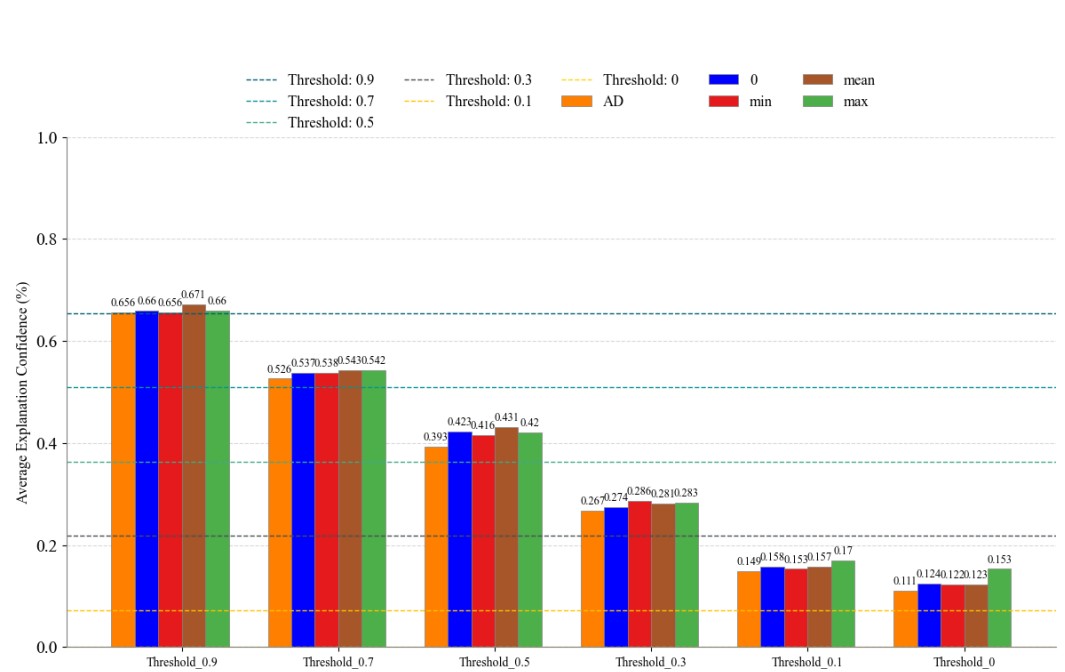

Figure G5: Average Explanation Confidence of different methods for RegNet-Y on ImageNet-1k

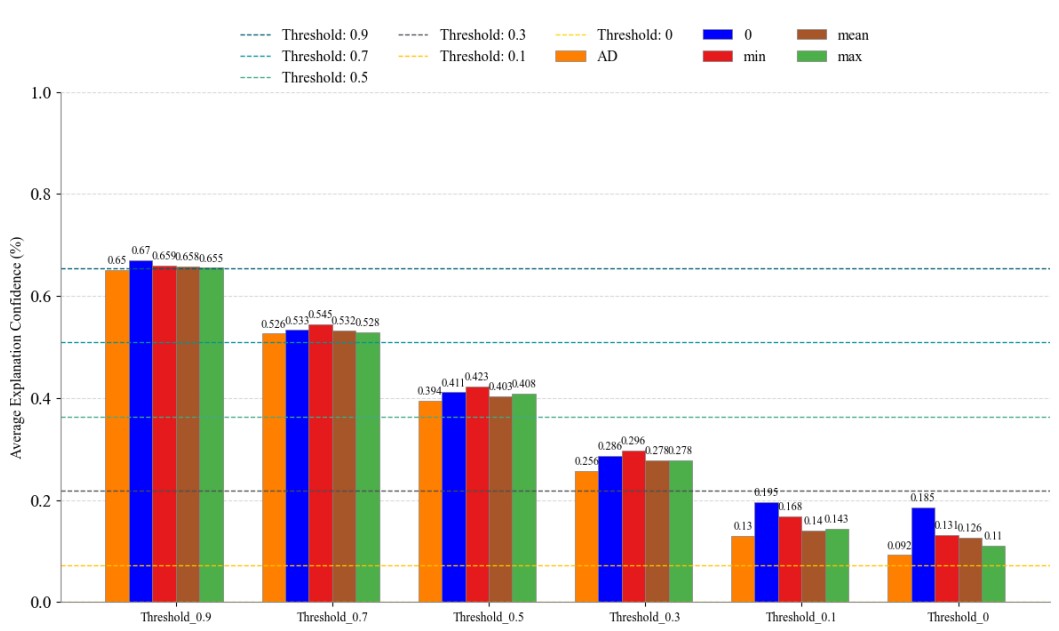

Figure G6: Average Explanation Confidence of different methods for RegNet-Y on IN-1k_V2

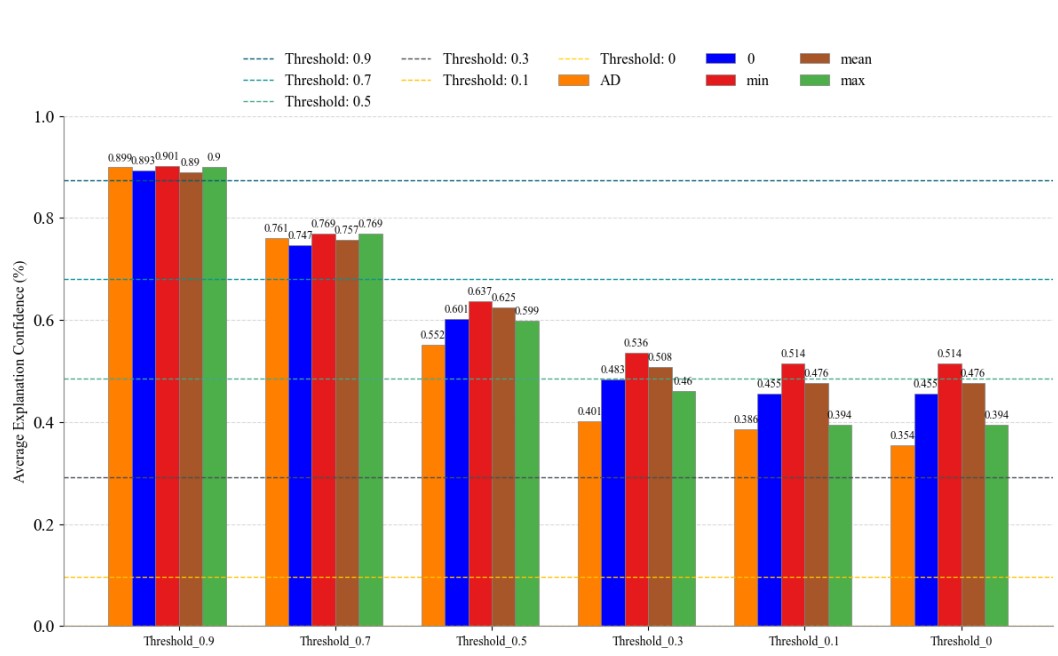

Figure G7: Average Explanation Confidence of different methods for RegNet-Y on CalTech-256

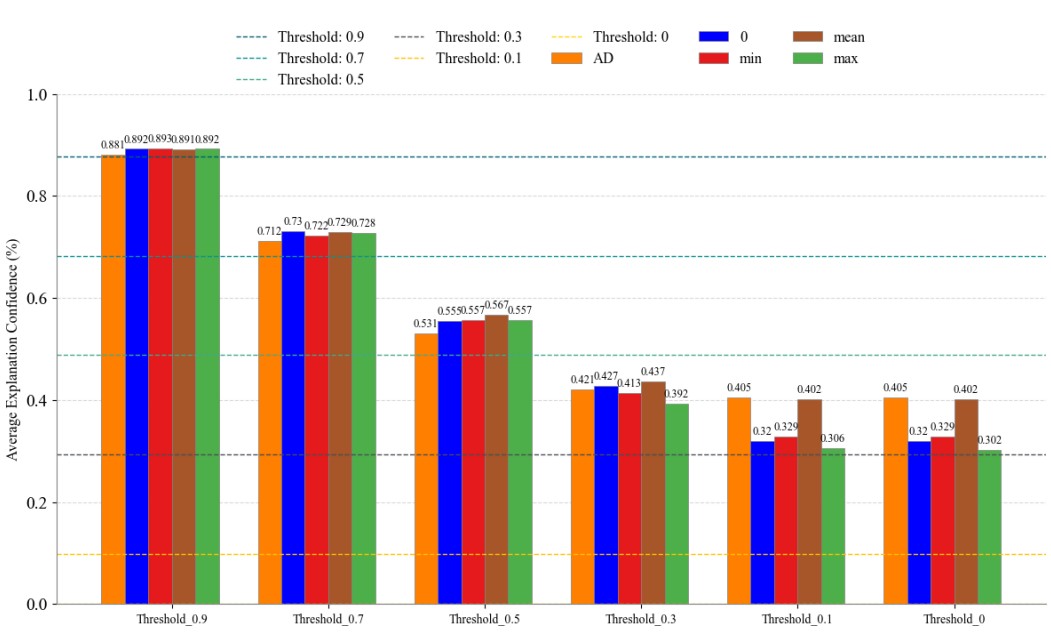

Figure G8: Average Explanation Confidence of different methods for RegNet-Y on PASCAL-VOC

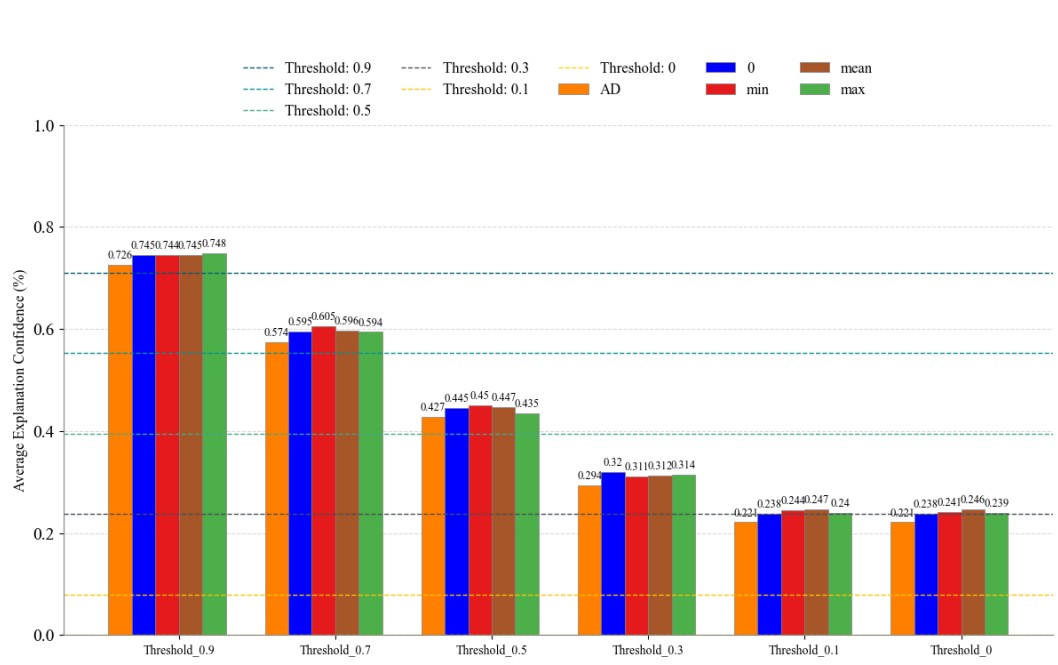

Figure G9: Average Explanation Confidence of different methods for EFN-V2 on ImageNet-1k

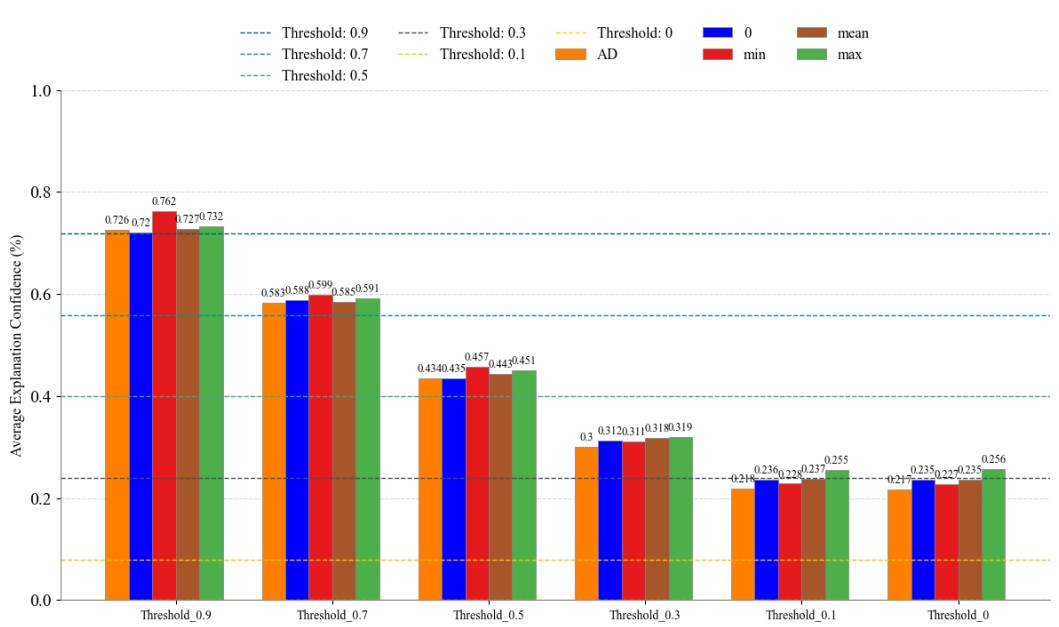

Figure G10: Average Explanation Confidence of different methods for EFN-V2 on IN-1k_V2

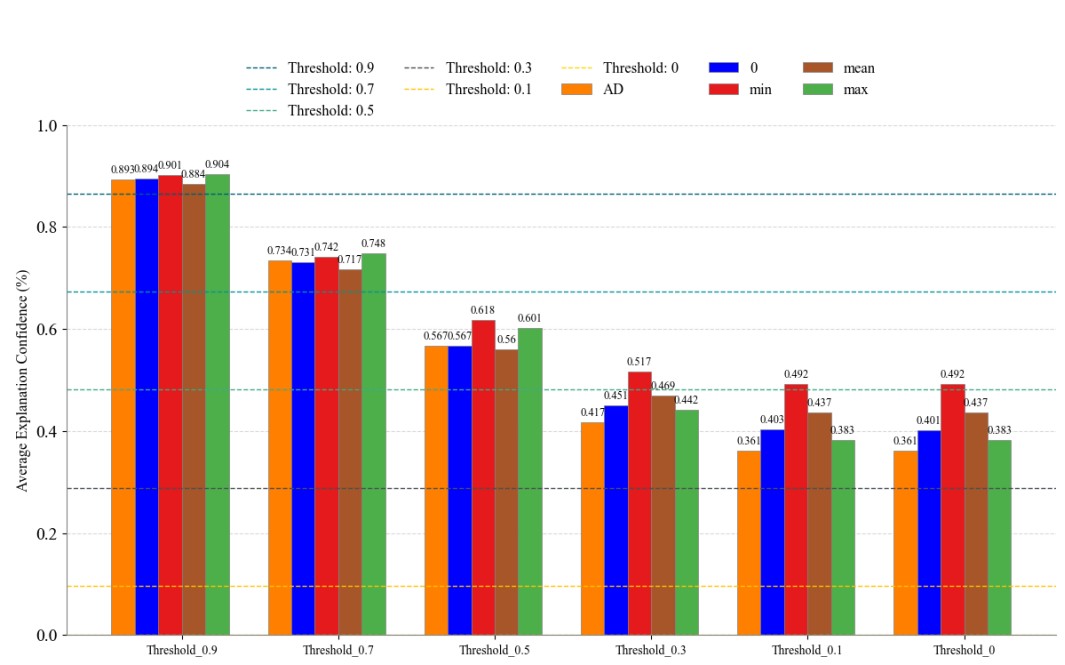

Figure G11: Average Explanation Confidence of different methods for EFN-v2 on CalTech-256

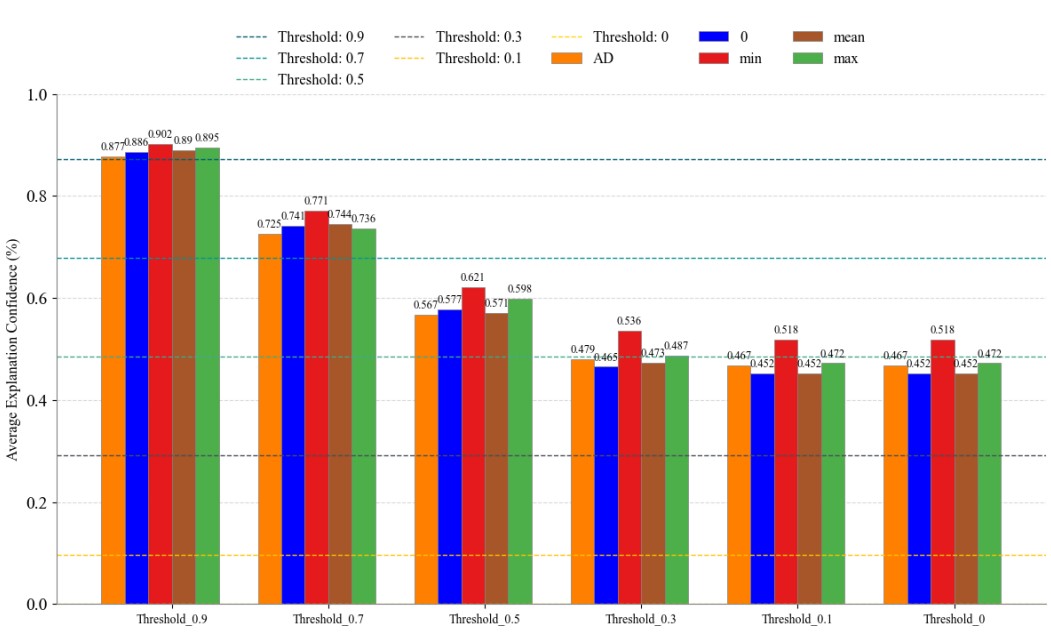

Figure G12: Average Explanation Confidence of different methods for EFN-v2 on PASCAL-VOC

