# OpenReview forum: "Activation-Deactivation: General Framework for Robust Post-hoc Explainable AI"
_ICLR.cc/2026/Conference — Submitted to ICLR 2026_

### Official Review · Reviewer_QVJX · 2025-10-18

**Soundness:** 2
**Presentation:** 2
**Contribution:** 2
**Rating:** 4
**Confidence:** 4

**Summary:**

The paper proposes a method called Activation-Deactivation to address the issue of OOD masked examples in mask-based explanation methods. This novel paradigm forces the model to ignore occluded parts of the input by deactivating activations at each layer of the model that correspond to the occluded features. By doing so, it halts the propagation of effects to subsequent layers. Experimental results show that the proposed method improves the robustness of explanations compared to those obtained using occlusions with min, max, avg, or zero pixel values.

**Strengths:**

1.	Critical Issue Addressed: The paper tackles the significant issue of OOD inputs caused by directly masking the input. Such inputs often fail to reflect the model’s decision-making process accurately, which is a critical challenge for black-box explainers.

2.	Theoretical Support: The methodology is theoretically grounded, with theorems and proofs provided.

**Weaknesses:**

1. While the method aims to address issues in black-box explainers, the Activation-Deactivation process requires access to the model architecture. This effectively transforms the method into a white-box explainer, raising concerns about evaluation fairness since other baselines in the comparison do not have access to the model architecture.

2. The evaluation section lacks clarity and detail, leading to confusion.

    2a. K1 and K2 are mentioned in Section 5.1, but only results related to K1 are reported. If K2 (images from a different class in the same dataset) is not evaluated, why is it mentioned?

   2b. The process of building IID backgrounds is not described in sufficient detail.

    2c. The robustness evaluation process is unclear. Based on the paper, my understanding is that given solid-colored images as backgrounds, the original image pixels are inserted into the background according to ranks identified by the explainer. If this is correct, more description of the evaluation process is necessary.

3. Generality: The method is limited in scope. It is only conducted and evaluated on CNNs and a single black-box explainer, REX. Its generalizability to other vision models (e.g., ViTs) and other black-box explainers remains unknown.

4. Lack of Visual Examples and User Study:

    The paper provides only one visual example, which is insufficient for an XAI method for image classification.

    No user study is conducted to evaluate interpretability for human users, which is a key aspect of XAI as it aims to help users understand the model's decision-making process.

5. The paper does not discuss or compare its method with prior related works, such as:

    	Explaining by Removing: A Unified Framework for Model Explanation (JMLR)

    	On the Robustness of Removal-Based Feature Attributions (NeurIPS 2023)

    	Can We Faithfully Represent Masked States to Compute Shapley Values on a DNN? (ICLR 2023)

**Questions:**

See weaknesses above.

---

> ### Author Response · Authors · 2025-11-13
> **Re: black-box or white box method?**
>
> The explainability process is still black-box. The network is modified to enable AD. In some sense, the network becomes "inherently explainable". Once this modification to the network is done, the algorithm is entirely black box.

---

> ### Author Response · Authors · 2025-11-13
> **Re: lack of user study**
>
> The response here is repeated from our response to reviewer mj3E.
>
> The example of ibex is for illustration only, showing that ConvAD produces semantically meaningful explanations, and is not intended to be a part of the evaluation. Human understandability is not the goal of the paper, nor do we claim that it is always true. To cite the recent paper "FACE: Faithful Automatic Concept Extraction" (NeurIPS'25), "This highlights the importance of aligning explanations with model behavior rather than visual plausibility alone. After all, models are not constrained to use human-understandable cues; they only use features that minimize loss." We do not use human understandability as a criterion for determining the quality of explanations - our goal is instead to determine the model's behavior that led to the classification, without the potential effect of masking values.

---

### Official Review · Reviewer_mj3E · 2025-10-30

**Soundness:** 1
**Presentation:** 2
**Contribution:** 1
**Rating:** 2
**Confidence:** 4

**Summary:**

The paper introduces a mechanism called CONVAD that removes the effects of occluded input features typically used by feature attribution methods to estimate feature importance given a prediction. The idea is to modify the forward pass of a convolutional neural network by deactivating neuron outputs (i.e., setting them to zero) associated with occluded features. Starting from the mask produced by an occlusion method, the mechanism updates the mask at each step of the forward pass by considering the ratio of masked to unmasked regions within each neuron’s receptive field. The mechanism is tested over the ReX feature attribution method across three architectures and three datasets, comparing it with alternative ways to occlude the input. The authors evaluate the resulting ReX explanations in terms of robustness, size, and confidence changes induced in the model by the modified forward pass, finding that the explanations computed by their proposed mechanism are more robust than those produced by alternative occlusion strategies (i.e., occlusion values).

**Strengths:**

- To the best of my knowledge, the idea is novel
- In the specific case of ReX, the performance of the proposed method is more consistent across several datasets than with alternative occlusion values. Avoiding the problem of choosing the right occlusion value per dataset is a nice contribution.

**Weaknesses:**

1) There is a general **mismatch between the contributions and claims and what is currently demonstrated in the paper** regarding solving the OOD problem and providing guarantees about the decision-making process. Specifically:
   - The paper states that *“We prove that CONVAD mechanism does not change the decision-making process of the network”*. In this regard, the paper proves that when no masks are provided, the decision process is the same (Theorem 1), but the central use case of the paper is **when masks are provided as additional input, in which case there is no guarantee**. In fact, **the decision process, intended as the model’s response to a given input, is guaranteed to change** because the method actively modifies that response by setting neurons in previous layers to zero; this can be verified by comparing the activations of the untouched neurons before and after applying CONVAD to check whether they differ: if they differ, even marginally, then the decision process is different. The fact that the original and modified forward passes yield the same prediction (or the same confidence) does not guarantee that the decision process is the same, nor does it imply that the explanations must be the same.
   - Authors claim to solve the OOD problem. **The OOD problem pertains not only to the input but also to the network’s response**; as the authors observe, *“if the occlusion value carries semantic meaning for the current model and current input dataset, the results might not reflect the model’s reasoning on the original input”* and the same concern applies to any change in the network response: if the edit carries semantic meaning for the current model, the results might not reflect the model’s reasoning on the original input. In this regard, the proposed method modifies the decision process and does not provide any guarantee that the modified response is in-distribution with respect to the activation distributions during training

2) Limited Evaluation Setup (Generalization): The paper claims to “solve both the OOD and the occlusion value problems” of occlusion-based black-box interpretability methods (in general). However, the experimental setup **tests the proposed method only on explanations computed by the ReX framework**. Therefore, at the current stage, the proposed method can be seen as an improvement to the ReX framework rather than a more general paradigm to “obviate the need for occlusions in post hoc explainability.” If the authors would like to generalize their claim, they should provide evidence that, when applied to a wide range of masks produced by different explanation methods, the results are consistent.

3) Custom evaluation Setup (Metrics): There is a large body of literature on the evaluation of feature attribution methods and, more generally, on occlusion-based methods. The paper proposes its own evaluation setup by *“planting” explanations onto different randomly selected colors and background images*. In this regard, it is unclear how this setup differs from standard ones used in the literature, what its advantages are, and **why standard procedures and standard metrics used in the literature for feature attribution (e.g., fidelity/faithfulness/del scores, etc) cannot be used in this context**. For example, several notions of robustness already exist in the literature, as the authors note; it is not clear **why a new definition of robustness is needed** and why other definitions cannot be applied; in a few words, the paper lacks contextualization regarding the evaluation setup. In this context, it is unclear why the size of the area of the explanations and the change in confidence of the model should indicative of the quality of explanations.

4) The **“Related Work” section is limited** and does not properly contextualize the proposed method within the appropriate literature. There is little discussion or mention of the substantial body of work dealing with occlusions/perturbations for post hoc methods (e.g., meaningful perturbations) and the literature on the evaluation of similar methods. This point is connected to the previous ones.

5) There are **several sentences in the paper that are not clear or questionable**. Some examples are the following:
   - “It only needs to be performed after a set of parametrized operations which have downstream effects in calculating future intermediate representations (such as pooling, concatenation, convolution etc.)”
Which operations do not have downstream effects?
   - Authors state that *“For this image of an ibex, the AD explanation clearly matches our intuition, as it consists of the ibex’ head with its unique-looking horns. In contrast, explanations obtained using different common occlusion values show small areas of the background and contain almost no part of the ibex.”*
There is a large body of literature indicating that explanations need not align with human intuition but should align with model behavior, and the reverse can cause overtrust and misleading explanations; moreover, several papers show that models similar to those tested by the authors may exploit unrelated features (e.g., small background parts) for the predictions. Therefore the visual difference is not very indicative of (or does not guarantee) improvements in the explanation quality since it depends on the decision process of the model.
   - The paper states *“ While we have not performed a user study, and it is out of scope for this paper, larger and more robust explanations generated by CONVAD are likely to increase trust in both an explanation and a model.”* This point is connected to the evaluation setup. The claim that larger explanations lead to more trust is not supported by evidence in the paper. For example, an explanation that covers the full input does not necessarily lead to more trust. Furthermore, “more trust” can lead to overtrust and be detrimental to the explanation process

6) Corollary 3.2 is stated as a consequence of a paper that, at the best of the reviewer’s knowledge, is a pre-print and relatively new (July 2025 on arxiv). While it doesn’t have direct consequences on the paper, I believe that at least the **proofs/corollary/theorems should be based on peer-reviewed papers**.

7) Results do not report errors bar or **statistical significance**. Therefore it is difficult to assess the validity of such results

**Questions:**

See weaknesses.

An additional (minor) question is the following:
- The authors state: *“Our results are over CNNs: it is straightforward, however, to extend AD to other architecture classes.”* Can the authors provide a description of extension of AD to an architecture that does not use receptive fields and in which each neuron is connected to all neurons in the previous layer? From my understanding, this method seems tailored to CNNs, which is fine, but extending it to other architectures would require alternative projection methods.

---

> ### Author Response · Authors · 2025-11-13
> **Re: change in the decision-making process as a result of the AD procedure**
>
> The reviewer points out that with masking, the ConvAD algorithm does not guarantee that there is no change in the decision-making process of the network. This is indeed correct. Our Theorem 1 proves the equivalence for the case where the masks are not provided (the base case). We will make it more clear in the paper - will update it right away. Note, however, that even with masking, the ConvAD mechanism does not create new pathways in the network - rather, it removes some pathways. So, while the decision process can change with the masking, the updated decision process was also possible before the masking.  Using the AD method with ReX ensures that we find a minimal subset of the input that is sufficient for the classification, when the model does not see the rest of the image (this is ensured by the AD).

---

> ### Author Response · Authors · 2025-11-13
> **Re: the OOD issue**
>
> The reviewer points out that the approach, while solving the OOD problem for the input, creates a potential OOD problem for the network’s response. Again, this is correct, and we are editing the paper to make it clear.  The question of which OOD is preferable is answered in the paper empirically, by comparing our explanations with the explanations obtained by occlusion-based methods. Wrt the possible semantic meaning - the AD process eliminates the concern for a semantic meaning of an occlusion value, by removing the need for occlusions. Note also that, while the original reasoning of the model on the input might have been different, the reasoning pathway that was found by ReX+AD was also possible before, and it results in a minimal sufficient subset for the classification.

---

> ### Author Response · Authors · 2025-11-13
> **Re: explanation of ibex being intuitive**
>
> The example is for illustration only, showing that ConvAD produces semantically meaningful explanations. This is not the goal of the paper, nor do we claim that it is always true. To cite the recent paper "FACE: Faithful Automatic Concept Extraction" (NeurIPS'25), "This highlights the importance of aligning explanations with model behavior rather than visual plausibility alone. After all, models are not constrained to use human-understandable cues; they only use features that minimize loss." We do not use human understandability as a criterion for determining the quality of explanations - our goal is instead to determine the model's behavior that led to the classification, without the potential effect of masking values.

---

> ### Author Response · Authors · 2025-11-13
> **Re: using results from unpublished papers**
>
> While Corollary 3.2 cites a paper that is available on arXiV, we will restate the theorem and provide its proof in the appendix.

---

> ### Comment · Reviewer_mj3E · 2025-11-26
>
> Thank you for the provided explanations. May I ask whether the revised version is complete, and request that it be uploaded with the modifications highlighted in a different color? This would facilitate a clearer visualization of the progress made. Since the rebuttal states that several parts will be changed, it is difficult to evaluate the progress without access to the revised version. Thank you.

---

### Official Review · Reviewer_ZAUp · 2025-11-03

**Soundness:** 2
**Presentation:** 2
**Contribution:** 2
**Rating:** 4
**Confidence:** 3

**Summary:**

The paper proposes Activation–Deactivation (AD), a forward-pass explanation method that keeps the original image but selectively disables internal feature locations tied to masked regions, avoiding the out-of-distribution (OOD) problem of input occlusion. Implemented as ConvAD, this drop-in module for CNNs yields more robust explanations than standard occlusion baselines on ImageNet-1k, ImageNet-v2, Caltech-256, and PASCAL-VOC, while leaving the model’s original predictions largely intact.

**Strengths:**

1. The paper targets a real, widely acknowledged issue in XAI for vision: explanations that depend on masked or patched images can be unreliable because the model is evaluated on OOD inputs. AD sidesteps this by keeping the input in-distribution and instead constraining network to ignore masked regions.

2. ConvAD is explicitly designed to be inserted into pretrained CNNs, with a proof/argument that an all-ones mask recovers the original forward pass. This makes the method appealing for practitioners who cannot retrain large models.

3. The authors test multiple CNN backbones and several datasets, and show that the robustness improvements are consistent across different confidence thresholds. This supports the claim that the method is not tightly coupled to a single dataset.

**Weaknesses:**

1. All experiments are done on CNN classifiers, while the paper presents AD as a general mechanism. Given the prevalence of vision transformers and CLIP-style models, at least a small-scale replication on a ViT or hybrid backbone would strengthen the generality claim.

2. The method is mainly compared to occlusion with different baselines, but not to strong, widely used methods (CAM-based XAI methods, SHAP, prototype-based XAI). Without these, it is hard to position the contribution against established baselines.

3. The paper reports its own robustness-style metrics (re-planting explanations on backgrounds, confidence preservation), but does not report standard faithfulness metrics such as Insertion/Deletion AUC or Average Drop / Increase in Confidence, which would make the results directly comparable to prior XAI work.

**Questions:**

N/A

---

### Author Response · Authors · 2025-11-13
**Re: the question of generalization to other architectures (namely, transformers)**

Wrt the proposed extension to the AD framework, we would like to point out that the paper introduces the general AD framework and an algorithm and its implementation for CNNs. We simply don’t have space to add the transformers-based algorithm to the same paper, so we defer it to the future work.

---

### Author Response · Authors · 2025-11-13
**Re: the lack of experiments with other XAI tools**

The AD-framework specifically targets XAI approaches, which rely on occlusions or perturbations in order to calculate attribution. CAM-based XAI methods and prototype-based XAI methods do not fall under this category.
Black-box XAI tools, such as SHAP, are within the scope of methods that AD addresses. However SHAP and similar methods do not include an extraction procedure of minimal sufficient sets and thus do not allow for comparison of the minimal sufficient sets natively. The only possible comparison we can make with SHAP is comparing the saliency maps, which provide some information about which input features are important for the output, but this information is not formalized or quantified.
To extract minimal sufficient sets from saliency maps, we need to add an extraction procedure. We will do that and report on results for a selection of other occlusion-based methods.

---

### Author Response · Authors · 2025-11-13
**Re: introduction of a new robustness metric**

Note that we are comparing sufficient approximately minimal subsets of the input. Commonly used metrics such as Insertion/Deletion AUC or confidence increases are not appropriate (as the subset is already close to minimal). This is why we introduce new metrics that compares robustness of the explanations as a proxy for their quality.

---

### Author Response · Authors · 2025-12-02
**a revised version of the paper with changes highlighted**

Following the reviewers' request, we uploaded a revised version of the paper with the changes highlighted in the pdf.

---

### Author Response · Authors · 2025-12-02
**Results for AD with LIME**

Note that the revised version also includes the results of running the AD mechanism with LIME. These are presented in Figure 6, which is a new figure (also highlighted in the pdf).

---

### Meta-Review · Area_Chair_hZjY · 2025-12-26

**Summary:**

The reviewers agreed the objectives of the paper, i.e. addressing the incorrect practice of evaluating explanation methods on OOD inputs, to be highly relevant.

A major shared concern included the claim of the general applicability of the proposed method while it seems the proposed method is tailored to CNN-type architectures. Given the prevalence that transformer architectures have nowadays, experiments on these architectures are a must to ensure a proper insight on the capabilities of the proposed method is provided.

Another shared major concern was on the empirical validation of the proposed method, where standard explanation method families and evaluation protocols/metrics from the literature have been ignored to a good degree. Moreover, as accurately pointed by Reviewer QVJX, the only considered competing (perturbation/occlusion -based) methods do not have access to the internal of the methods which clearly puts them in disadvantage w.r.t. the proposed method (which does have access to such information) and raises questions on the fairness of the comparison.

Very related to the above, several reviewers (Reviewer mj3E & QVJX) flagged that the proposed methods was not properly positioned with proper counterparts from the literature.

While the authors responded to some of the raised issues, the provided feedback acknowledged the existence of these issues. In other cases the provided answers were not as conclusive as to address these issues. Consequently, grounds for a recommendation towards the acceptance of this manuscript are limited.

**Reviewer Concerns:**

Addressed Concerns:

- While a response was provided to some of the raised concerns, in many cases the authors acknowledge the existence of the issues behind the concerns.  In other cases the provided feedback was not sufficiently conclusive as to properly address the concerns.


Outstanding

- Reviewer ZAUp:

    - Experiments exclusive to CNN classifiers.

    - Non-representative coverage of existing methods in the literature.

    - Dismissal of existing evaluation protocols/metrics.

- Reviewer mj3E

    - Mismatch between contributions and claims

    - Limited generalization of the adopted evaluation setup.

    - Dismissal of standard evaluation protocols.

    - Limited Related work section.

    - Unclear/questionable statements.

- Reviewer QVJX

    - White-box properties of the proposed method and unfairness of the evaluation.

    - Clarity of the evaluation

    - Applicability of the proposed method beyond CNN-based architectures.

    - Lack of visual examples and user study

    - Missing positioning w.r.t. related efforts.

**Reviewer Scores:**

The rebuttal was not conclusive enough as to trigger reviewers to increase their scores. A good indicator of this is the message from Reviewer ZAUp who explicitly indicated that after the rebuttal he/she would keep his/her original scores.

---

### Decision · Program_Chairs · 2026-01-26

Reject